# TECRL deficiency results in aberrant mitochondrial function in cardiomyocytes

Cuilan Hou[1,2,4], Xunwei Jiang[1,4], Han Zhang[1], Junmin Zheng[1], Qingzhu Qiu[1], Yongwei Zhang[1], Xiaomin Sun[1], Meng Xu[1], Alex Chia Yu Chang [3✉], Lijian Xie [1✉] & Tingting Xiao [1,2✉]

Sudden cardiac death (SCD) caused by ventricular arrhythmias is the leading cause of mortality of cardiovascular disease. Mutation in TECRL, an endoplasmic reticulum protein, was first reported in catecholaminergic polymorphic ventricular tachycardia during which a patient succumbed to SCD. Using loss- and gain-of-function approaches, we investigated the role of TECRL in murine and human cardiomyocytes. Tecrl (knockout, KO) mouse shows significantly aggravated cardiac dysfunction, evidenced by the decrease of ejection fraction and fractional shortening. Mechanistically, TECRL deficiency impairs mitochondrial respiration, which is characterized by reduced adenosine triphosphate production, increased fatty acid synthase (FAS) and reactive oxygen species production, along with decreased MFN2, p-AKT (Ser473), and NRF2 expressions. Overexpression of TECRL induces mitochondrial respiration, in PI3K/AKT dependent manner. TECRL regulates mitochondrial function mainly through PI3K/AKT signaling and the mitochondrial fusion protein MFN2. Apoptosis inducing factor (AIF) and cytochrome C (Cyc) is released from the mitochondria into the cytoplasm after siTECRL infection, as demonstrated by immunofluorescent staining and western blotting. Herein, we propose a previously unrecognized TECRL mechanism in regulating CPVT and may provide possible support for therapeutic target in CPVT.

[1] Department of Cardiology, Shanghai Children's Hospital, School of medicine, Shanghai Jiao Tong University, Shanghai 200062, China. [2] NHC Key Laboratory of Medical Embryogenesis and Developmental Molecular Biology, Shanghai Key Laboratory of Embryo and Reproduction Engineering, Shanghai 200062, China. [3] Department of Cardiology and Shanghai Institute Precision Medicine, Ninth People's Hospital, School of medicine, Shanghai Jiao Tong University, Shanghai 200125, China. [4] These authors contributed equally: Cuilan Hou, Xunwei Jiang. ✉email: alexchang@shsmu.edu.cn; naijileix@aliyun.com; ttxiao2017@163.com

Sudden cardiac death (SCD) due to ventricular arrhythmias is the leading cause of mortality in patients with heart disease[1]. Catecholaminergic polymorphic ventricular tachycardia (CPVT), a prevalent cause of SCD during childhood and adolescence, is characterized by ventricular tachycardia or syncope due to emotional overstimulation or physical exercise[2]. CPVT is a life-threatening genetic disorder with an estimated prevalence of 1:10,000 in the general population[2]. Current treatments for CPVT include exercise restriction, β-blocker, class 1c sodium channel blockers, cardiac sympathetic denervation, and implantable cardiac defibrillators[3]. The treatments mentioned above can delay and reduce the incidence of SCD, but it still needs more effective therapeutic options to increase the quality of life in treating SCD.

Since 1970, several CPVT mutations in ion channel genes or calcium-binding proteins have been identified: ryanodine receptor 2 (RYR2), calsequestrin 2 (CASQ2), and triadin (TRDN)[4]. Recently, a mutation in trans-2, 3-enoyl-CoA reductase-like (TECRL) gene, coded for endoplasmic reticulum (ER) protein, was first identified in CPVT patients[5]. Our group identified a compound heterozygosity mutation in the TECRL gene (c.587 C > T and c.918 + 3 T > G) in a 13-year-old CPVT patient[6]. Even though the TECRL gene mutation has been identified with CPVT[5–7], the functions and pathogenic mechanisms of TECRL remain largely unknown and still need to be studied further.

TECRL encodes 363-amino acids and contains an n-terminal ubiquitin-like domain. The last two trans-membrane regions overlap with the 3-oxo-5-alpha steroid 4-dehydrogenase domain. TECRL mRNA is mostly expressed in the heart and skeletal muscles and is generally undetectable in other human tissues[5]. Using human-induced pluripotent stem cell-derived cardiomyocytes (hiPSC-CMs), TECRL deficient cardiomyocytes exhibited decreased calcium transient amplitude, elevated diastolic calcium, and increased Tau decay that was associated with abnormal SERCA and NCX activities[5].

Mitochondria account for 20–35% of the volume of a single cardiomyocyte and is central to energy production and metabolism[8]. Priori et al. first reported that Ryr2 mutated mice (R4496C) exhibited a reduction in the percentage of cardiac cells with severe mitochondrial abnormalities[9]. Miragoli et al. reported that displaced mitochondria due to myocardial infarction resulted in aberrant mitochondrial Ca$^{2+}$ propagation[10]. Mitochondria-ER structure plays a key role in myocardial mitochondrial homeostasis[11]. TECRL may have other regulatory roles in CPVT, besides the regulation of calcium transients[5]. We hypothesize that there may be a strong link between impaired TECRL signaling and the resultant mitochondrial dysfunction in cardiomyocytes, which may play a crucial role in CPVT disease.

To determine the role of Tecrl in mitochondrial homeostasis and cardiac function, we generated Tecrl-null (Tecrl KO) mice. Tecrl deficiency results in impaired cardiac function, altered cardiac metabolism, decreased mitochondria function, and elevated levels of reactive oxygen species (ROS). Mitochondrial fusion proteins such as MFN2 were decreased both in the Tecrl KO mice hearts and Tecrl- deficient cardiomyocytes (hiPSC-CMs). It is possible that TECRL overexpression can partially reverse MFN2 decline. Our results show that TECRL maintains mitochondrial homeostasis through regulation of MFN2, suggesting that TECRL may be a potential target for treating CPVT.

## Results

**Tecrl KO mice exhibit cardiac dysfunction.** To disrupt the open reading frame of Tecrl in mice, we designed a pair of sgRNAs targeting the murine Tecrl gene (Fig. 1a). The CRISPR/Cas9 system was utilized to produce Tecrl KO mice (Fig. 1a, b). Animals were genotyped using PCR (Fig. 1c), RT-PCR (Fig. 1d), and immunoblotting (Fig. 1e) to confirm the loss of Tecrl. And Tecrl was also not existed in the Tecrl KO mice (Table S1) through mice whole heart mass spectrometry, indicating that the Tecrl KO mice had been produced successfully.

To assess the cardiac function of the KO mice, we performed echocardiography on 8 weeks old WT and Tecrl KO mice using M-mode imaging (Fig. 2a). Compared to the WT mice, Tecrl KO mice exhibited significant contraction defects: decreased ejection fraction (LVEF), decreased fractional shortening (LVFS), increased left ventricular internal dimension systole (LVIDs), increased left ventricular internal dimension diastole (LVIDd), increased left ventricular end-systolic volume (LVESV), and increased left ventricular end-diastolic volume (LVEDV) (Fig. 2b). However, these changes were not observed in mice 4 to 5 weeks old (Fig. S1a, b). In the Tecrl KO mice, multiple premature ventricular beats and ventricular tachycardia (VT) were observed immediately after an epinephrine and caffeine (epi/caffeine) injection; however, few ventricular events were observed in WT mice (Fig. 2c, d, g). The premature ventricular beats (PVB) are shown in Fig. 2e. 60% of Tecrl KO mice suffer VT sustained for 1 to 5 s, 40% of them sustained 5 to 15 s, 30% of them sustained larger than 15 s, while none of WT mice suffered VT (Fig. 2f). Both the WT and Tecrl KO mice heart rates were increased significantly after epi/caffeine stimulation (Fig. 2h). Subsequent experiments were conducted in 8-week-old mice.

**Tecrl deficiency regulates the myocardial metabolism pathway.** To determine the impact of Tecrl deficiency on cardiac gene expression, we profiled the heart transcriptome of WT and Tecrl KO mice using RNA-seq. We identified 181 differentially expressed genes between the Tecrl KO and WT mice. Of these, 78 (43.1%) were upregulated and 103 (56.9%) were downregulated in the Tecrl KO mice compared to WT mice (Table S2). Using volcano plots, we noted that expression of Elovl3, Igf2, and Crlf1 were the most-robustly changed genes in the Tecrl KO mice compared to WT mice (Fig. 3a). To identify signaling pathways which were most affected by Tecrl deficiency, we performed gene ontology (GO) for enrichment analysis of differentially expressed genes. Here, we observed a disproportionate representation of genes involved in the rhythmic process, regulation of circadian rhythm, circadian rhythm, glycogen biosynthetic processes, and glucose metabolic processes (Fig. 3b) ($p < 0.05$). Similarly, KEGG pathway analysis indicated that mitogen-activated protein kinase (MAPK) and AMP-activated protein kinase (AMPK) signaling pathways were altered in the Tecrl KO hearts (Fig. 3c) ($p < 0.05$). Together, these data suggest Tecrl deficiency is associated with myocardial metabolism.

**Tecrl deficiency disturbs cardiac mitochondria and induces ROS levels.** Considering that cardiac dysfunction is closely associated with mitochondrial dysfunction, we first examined the mitochondrial ultrastructure using transmission electron microscopy (TEM). Tecrl KO murine hearts exhibited mitochondria ultrastructural changes, including irregular arrangement and loss of cristae (Fig. 4a). In Tecrl KO hearts, mitochondrial cross-sectional area and the percentage of mitochondria with disorganized cristae ultrastructure were significantly increased compared to WT hearts (Fig. 4b). Next, we assessed the mitochondrial respiratory capacity of 8-week-old adult cardiomyocytes using the seahorse assay. Compared to WT cardiomyocytes, Tecrl KO cardiomyocytes exhibited a significant decrease in mitochondrial basal respiration and ATP production (Fig. 4c, d). An increase in the maximum respiration trend was observed, but this was not statistically significant (Fig. 4c, d).

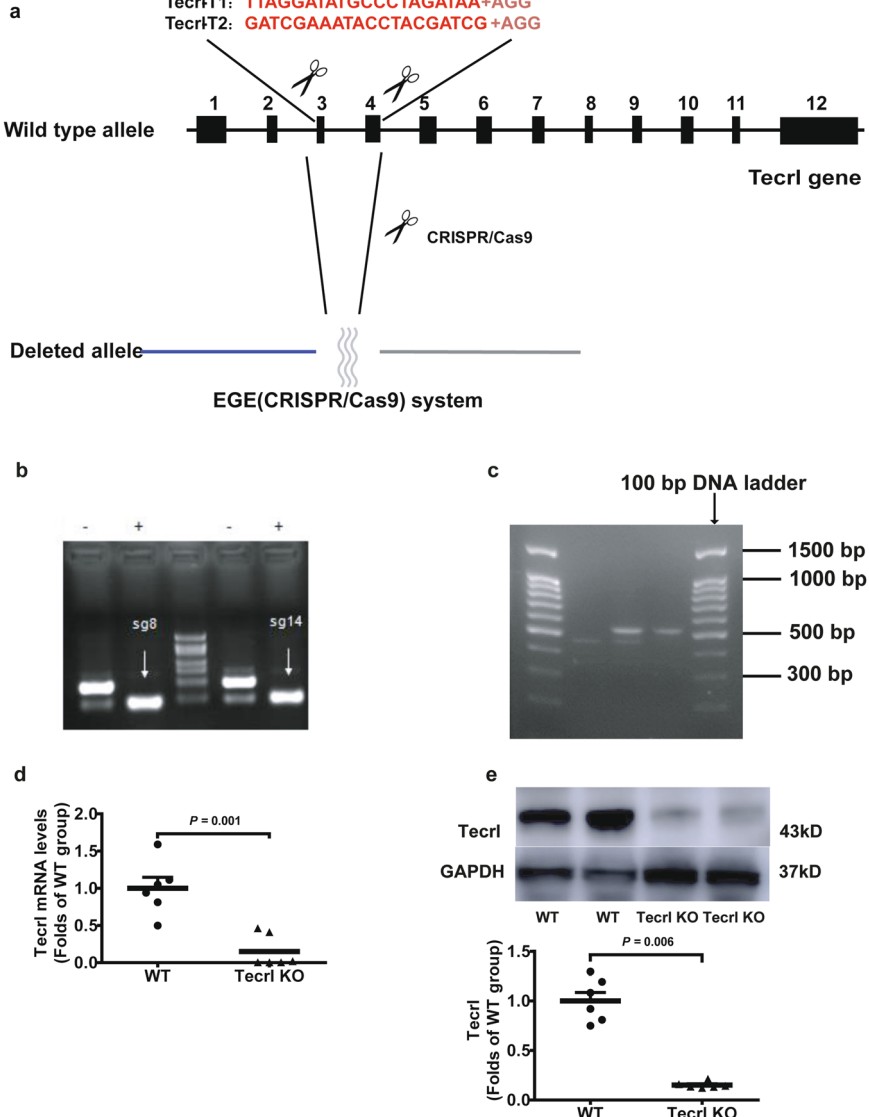

**Fig. 1 Generation of Tecrl KO mice using CRISPR/Cas9. a** The general structure of Tecrl gene. Two sgRNA sequences and a schematic diagram of two sgRNA target sites located in exon 3 and exon 4 of the mice Tecrl locus. Target sites of two sgRNA sequences (Tecrl-T1 and Tecrl-T2) are highlighted in red. **b** Representative of sgRNA images. **c** Representative of the PCR images. Only one band at position 537 represented the Tecrl KO mice, one band at position 455 represented the WT mice, while one at position 537 and one at position 455 represented the heterozygote mice. **d** Quantification of the Tecrl mRNA level in the WT and Tecrl KO mice. **e** Representative of the immunoblotting images and quantification of the expression of Tecrl in the WT and Tecrl KO mice. The DNA ladder (DL2000). Values are means ± SE. $P < 0.05$ was considered significant.

Mitochondrial dysfunction often results in an increase in ROS[12]. To quantify ROS levels, we used a dihydroethidium (DHE) fluorescence assay. DHE intensity was elevated in the 8-week-old Tecrl KO hearts compared to WT hearts (Fig. 4e, f). We also found that DHE and Mitosox levels were increased upon TECRL knockout in primary neonatal cardiomyocytes (Fig. 4g–j). Interestingly, DHE fluorescence intensity levels showed no significant difference between the WT and Tecrl KO groups when assayed in mice 4 to 5 weeks old (Fig. S2a, b). These data demonstrated that long-term Tecrl deficiency may induce an oxidative burden that is correlated with mitochondrial dysfunction.

**Tecrl deficiency regulates the cardiac mitochondrial proteome.** To determine the impact of Tecrl deficiency on mitochondrial protein composition, we performed proteomic analysis on purified mitochondria isolated from the Tecrl KO mice and WT hearts. We identified a total of 1891 proteins that differed between the Tecrl KO and WT hearts. Using a 1.3-fold cutoff and an FDR < 0.01, we found that 263 (85.9%) were upregulated and 43 (14.1%) were downregulated in the Tecrl KO mice compared to WT mice (Table S3). As shown in Fig. S3a, we noted that cytochrome P450 and D-lactate dehydrogenase were the most-robustly decreased proteins (FDR < 0.01) in the Tecrl KO mitochondria compared to WT mitochondria. Using KEGG enrichment analysis, we observed that these altered proteins were involved in metabolism, oxidative phosphorylation, fatty acid degradation, and arrhythmogenic right ventricular cardiomyopathy (ARVC), which were significantly decreased in the Tecrl KO mice compared to WT (Fig. S3b).

**Tecrl regulates mitochondrial function through PI3K/Akt signaling.** The regulation of the phosphoinositide-3-kinase (PI3K)/Akt murine thymoma viral oncogene homolog (Akt), MAPK, and AMPK pathways, all play essential roles in cardiovascular

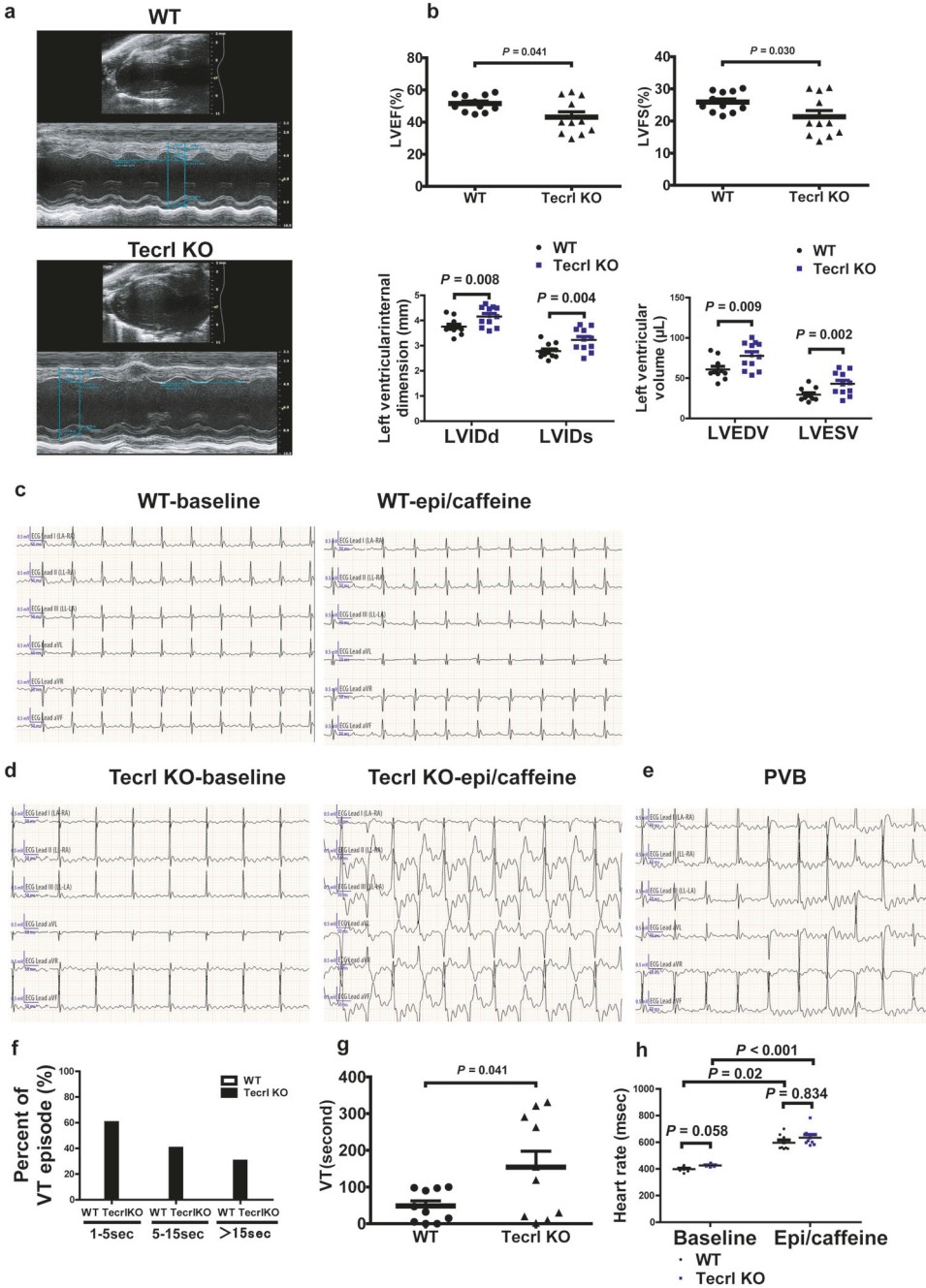

**Fig. 2 Tecrl deficiency-induced cardiac dysfunction. a** Representative echocardiography images of WT and Tecrl KO mice. **b** Quantification of LVEF, LVFS, LVIDd, LVIDs, LVEDV, and LVESV in WT and Tecrl KO mice (8 weeks) (*n* = 8). **c–f** Representative ECG recordings and percent of VT episode (%) in WT and Tecrl KO mice before and after epi/caffeine stimulation (*n* = 10). **g** Quantification of VT (*n* = 10). **h** Quantification of mice heart rate (*n* = 8). Values are mean ± SE. *P* < 0.05 was considered significant.

homeostasis[13,14]. To determine the relationship between the Tecrl and PI3K/Akt signaling during mitochondrial homeostasis, we measured the expression of mitochondrial fusion regulators: mitofusin1 (Mfn1), mitofusin2 (Mfn2), nuclear factor erythroid 2-related factor 2 (Nrf2), and PI3K/Akt. Compared to WT hearts, the Tecrl KO hearts exhibited a decrease in Mfn2, p-Akt (Ser473), and Nrf2; an increase in Fas; and a stable level of Mfn1 protein, which was measured by immunoblotting (Fig. 5a–d).

Next, we used hiPSC-CMs to validate whether the TECRL-PI3K/Akt signaling axis is evolutionary conserved. Human-induced pluripotent stem cells (hiPSCs), which stained positive for pluripotent marker OCT4 (Fig. 6a), were differentiated into

beating cardiomyocytes using an established protocol[15]. As shown in Fig. 6b, these hiPSC-CMs expressed α-actinin and cTnT, and spontaneous beating was observed (Video S1), indicating that these cells were hiPSC-CMs. The baseline beating frequency of hiPSC-CMs was measured. Statistical analysis showed no significant difference between the Ctl and shTECRL group (Table S4). Musclemotion was used to determine the velocity of contraction and relaxation of the hiPSC-CMs beating in a two-second-long video. Then we compared the hiPSC-CMs speed of contraction after TECRL knockdown and found that there was no significance between the Ctl and shTECRL group (Fig. 6c). Using shRNA knockdown, we observed that loss of

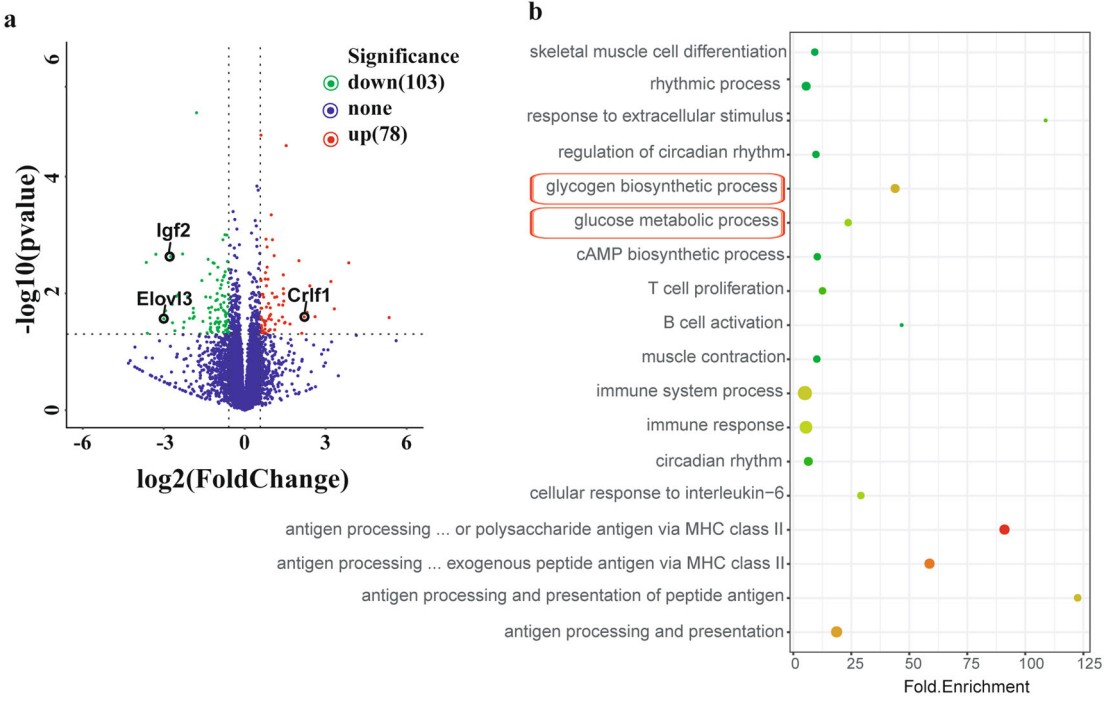

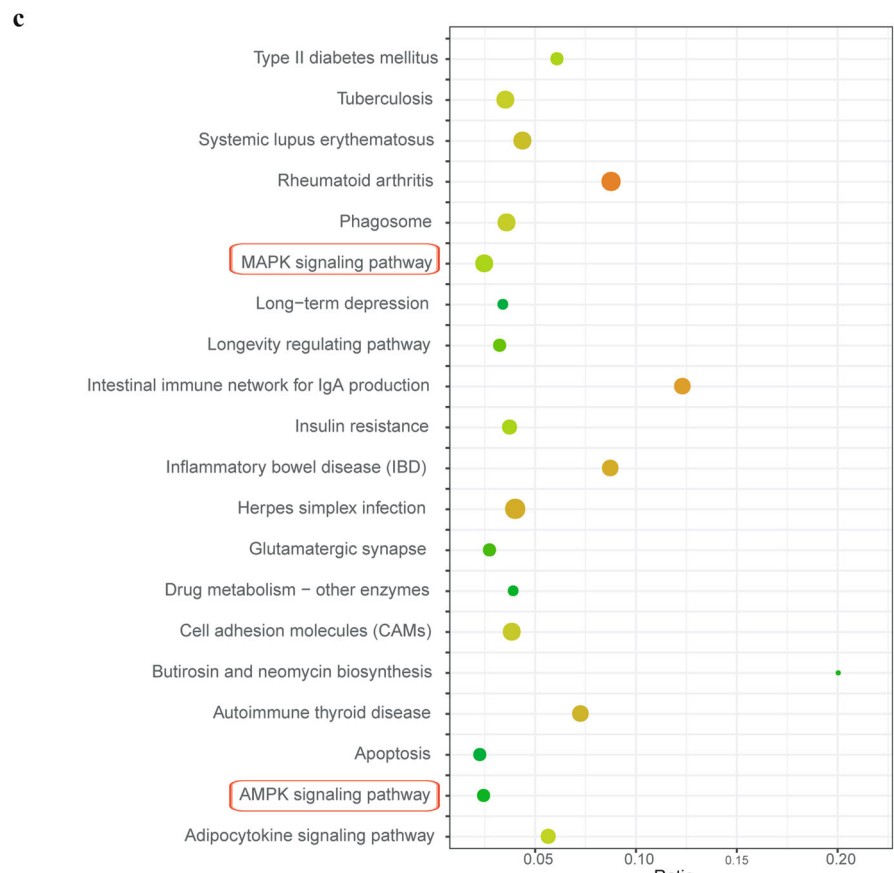

**Fig. 3 Visualization of the differentially regulated mRNAs in cardiac tissues of WT and Tecrl KO mice. a** Representative volcano plots of differentially regulated mRNAs in the WT and Tecrl KO mice cardiac tissues. Those with $P < 0.05$ and |fold change| >1.5 are highlighted. **b** GO analyses of differentially expressed mRNAs. **c** KEGG analyses of differentially expressed mRNAs ($n = 6$).

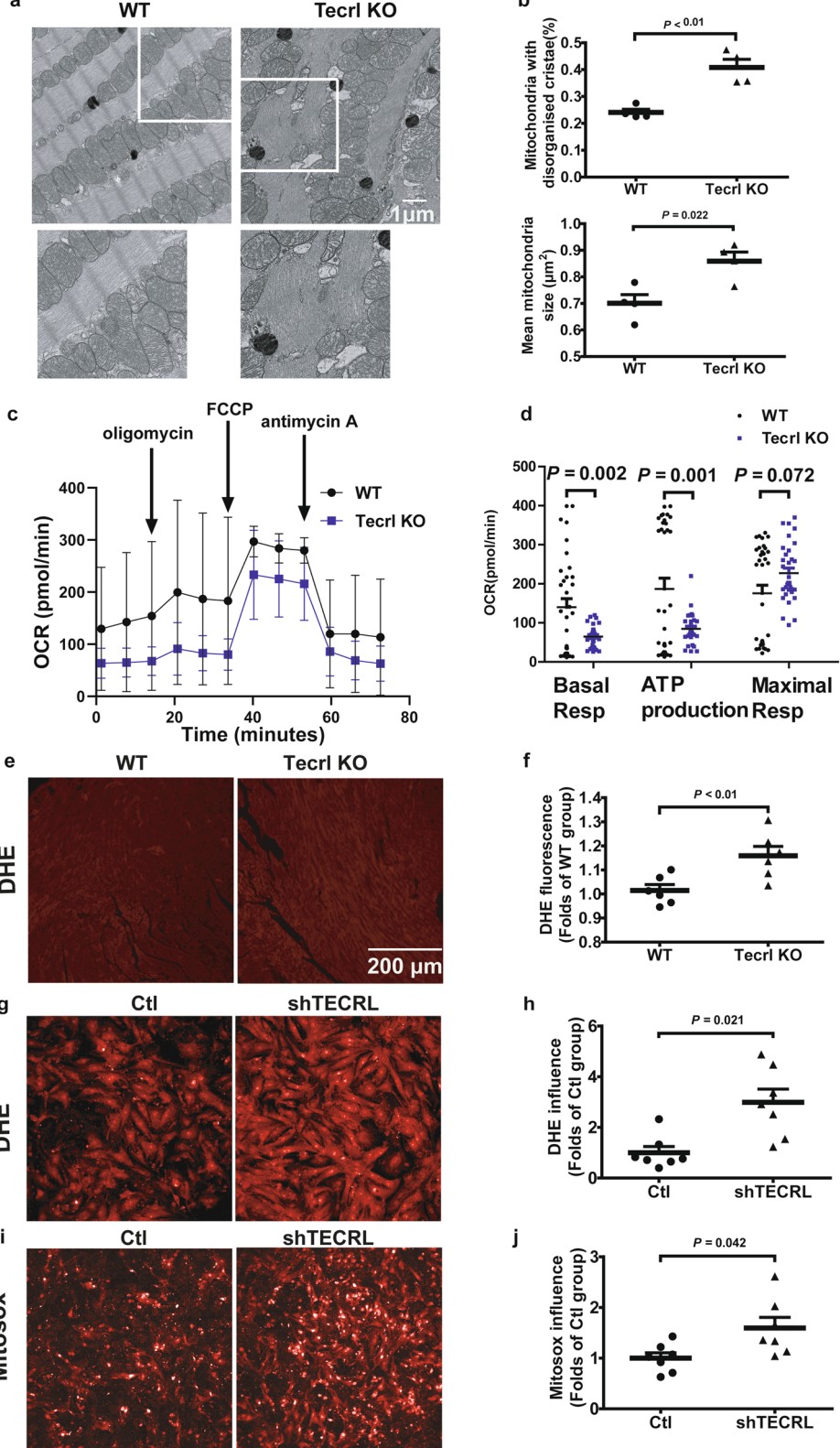

**Fig. 4 Tecrl deficiency significantly suppressed mitochondrial respiration. a**, **b** The cardiac ultrastructure of the mice was examined by TEM. Representative image and quantitative analysis of cardiac ultrastructure of the WT and Tecrl KO mice at the age of 8 weeks. Magnification is 6700 x, scale bar = 1 μm (n = 4). **c**, **d** Measurements of OCR and respective quantitative analysis of cardiomyocytes isolated from WT and Tecrl KO mice (WT, n = 34, Tecrl KO, n = 32). **e**, **f** Representative image and quantitative analysis of DHE staining. The WT and Tecrl KO mice tissues were both measured at the age of 8 weeks (n = 6). **g**, **h** Representative image and quantitative analysis of DHE staining in primary neonatal cardiomyocytes (n = 7). **i**, **j** Representative image and quantitative analysis of Mitosox staining in primary neonatal cardiomyocytes (n = 4). FCCP, trifluoromethoxy carbonyl cyanide phenylhydrazone. Values are mean ± SE. P < 0.05 was considered significant.

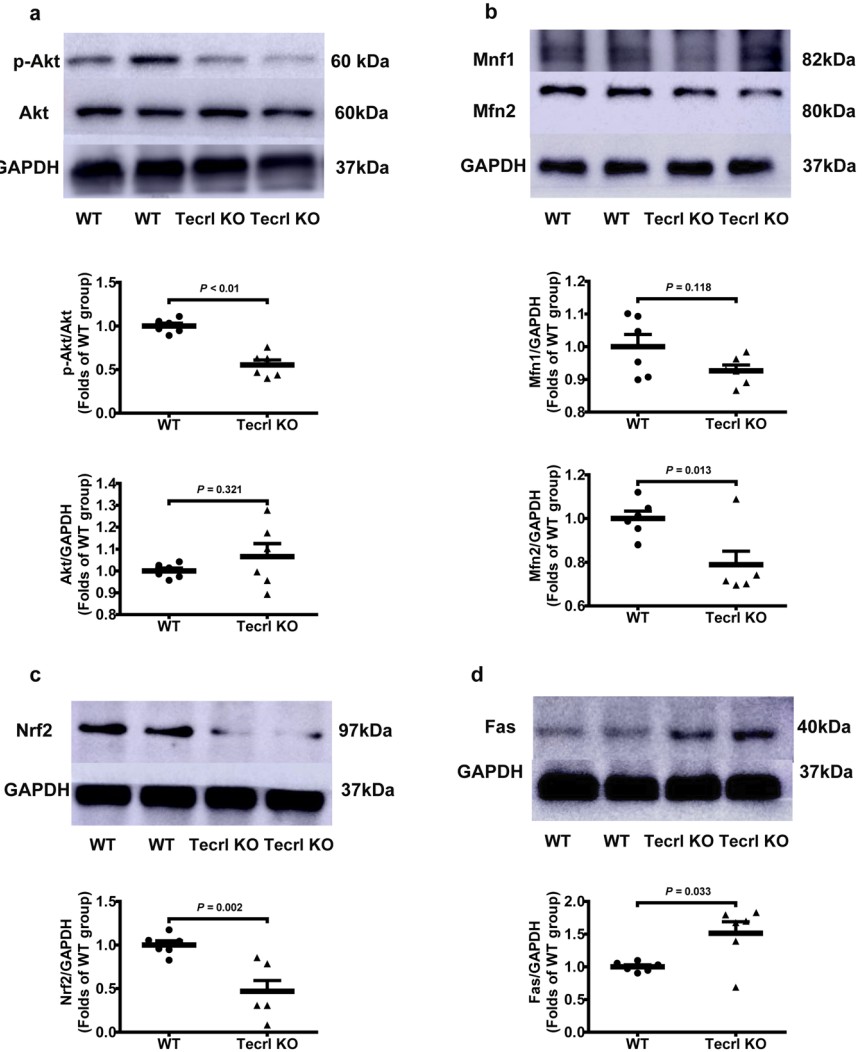

**Fig. 5 Tecrl regulates mitochondrial function through PI3K/Akt signaling and mitochondrial fusion protein MFN2. a–d** Representative immunoblotting images and quantification of the expression of p-Akt, Mfn1, Mfn2, Nrf2, Fas, and Tecrl in WT and Tecrl KO mouse hearts ($n = 6$). Values are mean ± SE. $P < 0.05$ was considered significant.

TECRL resulted in a decrease in protein levels of MFN2, p-AKT (Ser473), and NRF2 (Fig. 6d–g). We also confirmed that the mRNA (Fig. S4a–d) and protein (Fig. S5a–c) levels of MFN2 and NRF2 were decreased, via FAS increased in H9C2 cells. Next, we overexpressed TECRL in hiPSC-CMs, and mass spectrometry was performed. The mass spectrometry results showed that FAS decreased significantly upon TECRL overexpression (Table S5). Results of immunoblotting confirmed that FAS protein decreased following TECRL overexpression (Fig. 7a), which was in agreement with our findings in mice (Fig. 5d). Moreover, MFN2 was partially increased following TECRL overexpression (Fig. 7b). Based on the mass spectrometry of TECRL co-immunoprecipitation (Table S6), we observed that ATP synthase subunit beta, mitochondrial (ATPB) was in contact with TECRL, and the contact was significantly increased after TECRL overexpression (Fig. 7c).

HiPSC-CMs and H9C2 cells were utilized to measure mitochondrial respiration. Maximum mitochondrial respiration was significantly enhanced after TECRL overexpression in hiPSC-CMs, but was blocked after PI3K/Akt inhibitor LY294002 treatment (Fig. 7d, e). Similarly, mitochondrial respiration in H9C2 cells was significantly enhanced (basal respiration, ATP production, and maximal respiration) upon TECRL overexpression, and this

increase was also PI3K/Akt dependent (Fig. S4e, f). Apoptosis-related genes were related in the above cardiac RNA-seq (Fig. 3c). We further noticed that apoptosis-inducing factor (AIF) and cytochrome C (Cyc) were released from mitochondria into the cytoplasm (60 nM siTECRL) via immunofluorescent staining (Fig. 7f) and western blotting (Fig. 7g) in H9C2 cells. The 60 nM siTECRL infection increased the expression levels of Cyc and AIF in the cytoplasm. There was no significance in mitochondrial Cyc and AIF levels in between 60 nM siTECRL infection group and its negative control (Fig. 7g). Altogether, our data demonstrated that TECRL can regulate mitochondrial respiration through PI3K/Akt signaling.

## Discussion
The Tecrl KO Mice exhibited significant heart contraction defects in 8 weeks, even though the mice hearts were normal at 4 to 5 weeks old. In the Tecrl KO mice, we demonstrated that Tecrl can regulate mitochondrial homeostasis through PI3K/Akt. We showed that in the absence of Tecrl, cardiomyocytes exhibited decreased Mfn2, p-Akt (Ser473), and Nrf2 protein expressions; there was also an increase in apoptosis-inducing factor (AIF) and cytochrome C (Cyc) released from the mitochondria into the

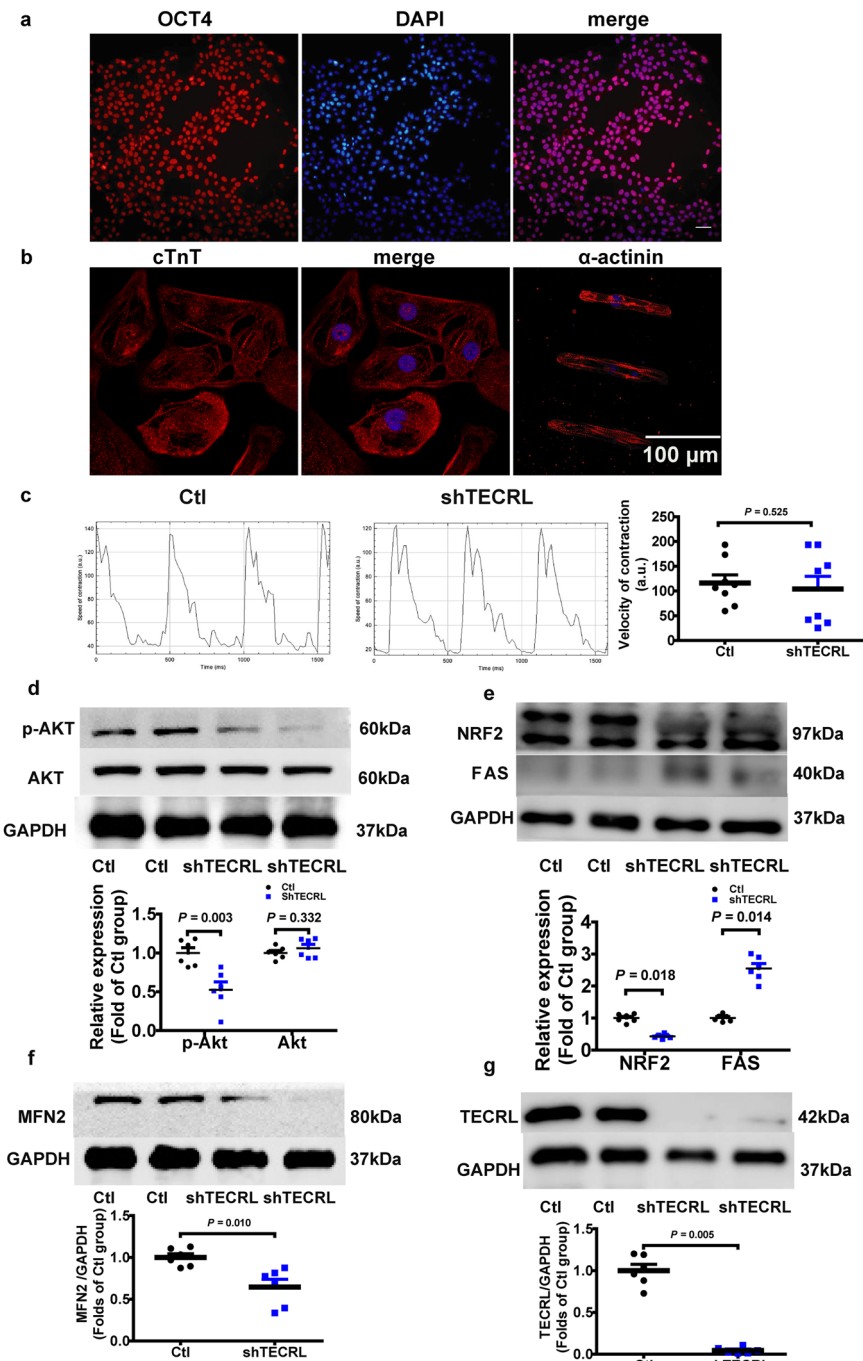

**Fig. 6 Tecrl regulates PI3K/Akt signaling and mitochondrial fusion protein in hiPSC-CMs. a** Representative immunofluorescence images of OCT4 in the hiPSCs ($n = 3$). **b** Representative immunofluorescence images of α-actinin and cTnT in the hiPSC-CMs ($n = 6$). **c** Representative images and quantification of the velocity of contraction and relaxation of hiPSC-CMs. **d–g** Representative immunoblotting images and quantification of the expression of p-AKT, NRF2, MFN2, FAS, and TECRL in hiPSC-CMs following TECRL knockdown. Values are mean ± SE. $P < 0.05$ was considered significant.

cytoplasm. These resulted in a decrease in mitochondrial respiration. Overexpression of TECRL enhances maximum mitochondrial respiratory capacity and this increase is in a PI3K/Akt-dependent manner.

Previous studies, conducted by our team[6] and others[5], reported that mutations in TECRL are linked to CPVT. And an international project to collate these findings has been launched[16]. CPVT patients are currently treated with conventional therapies, including β-blockers and channel blockers, which have limited efficacy in preventing arrhythmic events and SCD[17,18]. Currently, mutations in RYR2 and CASQ2 account for 50% and 5% of

CPVT cases respectively[19]. Mouse models with Ryr2 mutations (Ryr2[R4496C], Ryr2[N2386I], Ryr2[A165D])[20–23] and Casq2 mutations (Casq2[D307H], Casq2[DeltaE9/DeltaE9])[24] can effectively replicate the CPVT phenotype. Yet clinically, the prescription of channel blockers does not prevent SCD, suggesting an alternative pathogenic mechanism may be at play. It has been demonstrated that mitochondrial dysfunction may lead to calcium leakage[10], which can induce mitochondrial ROS production through an RYR2 mutation[25], further dampening mitochondrial function. TECRL deficient cardiomyocytes have reduced expression of RYR2 and CASQ2[5]. Here, we demonstrated that TECRL deficiency results in

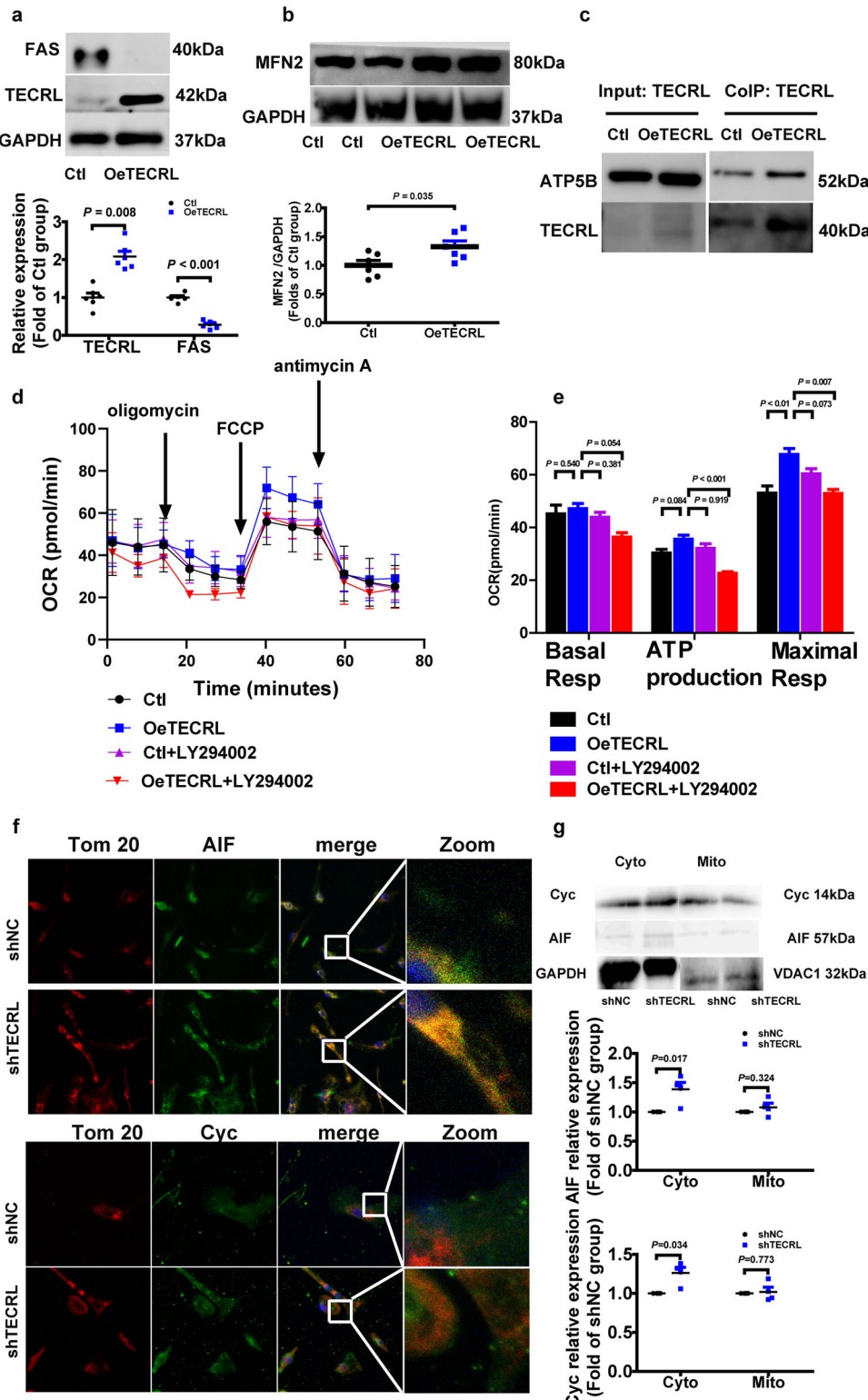

**Fig. 7 Overexpression of TECRL can induce mitochondrial respiration in hiPSC-CMs. a, b** Representative immunoblotting images and quantification of the expression of FAS, MFN2, and TECRL of hiPSC-CMs following TECRL overexpression (n = 6). **c** Representative images of co-immunoprecipitation of TECRL and ATPB (n = 3). **d, e** Measurements of OCR and respective quantitative analysis in hiPSC-CMs (n = 12). **f** Representative images of Cyc and AIF colocalization with Tom 20 in hiPSC-CMs (n = 6). Scale bar = 10 μm. **g** Cyc and AIF released from the mitochondria into the cytoplasm were analyzed by western blotting. VDAC1 was used as a loading control for mitochondria gradients. GAPDH was used for cytosolic gradient (n = 4). Values are mean ± SE. P < 0.05 was considered significant.

a late-onset mitochondrial dysfunction (mitochondrial dysfunction not evident in 4–5 weeks old).

Ultrastructural contacts between ER and mitochondria make up the mitochondria-associated ER membranes (MAM). MAM structures play a pivotal role in $Ca^{2+}$ signaling, lipid transport, energy metabolism, and cell survival[26,27]. Wu et al. found that hyperglycemia or high glucose-driven inactivation of AMPK results in aberrant MAM formation, mitochondrial $Ca^{2+}$ increase, mitochondrial dysfunction, and cardiac dysfunction[28]. Santulli et al. showed that 27 RYR2 CPVT patients displayed abnormal oral glucose tolerance and attributed this observation to sarcoplasmic reticulum stress-induced b-cell apoptosis[23]. TECRL expression is restricted to cardiac and skeletal muscles and is mostly localized to ER[5]. Bioinformatics analysis showed that the function of TECRL protein focused on very-long-chain fatty acid biosynthetic processes and oxidoreductase activity. We showed that loss of Tecrl results in altered Nrf2 as well as Fas expression (Figs. 5, 6). Given that adult cardiomyocytes utilize free fatty acids as their main metabolic substrate, mitochondrial dysfunction in TECRL deficient cardiomyocytes supports the notion of an alternative pathway driving CPVT[29]. Mitochondrial homeostasis is vital for mitochondrial health[30], and it has been demonstrated that disruption of mitofusion[31] and mitofission[32] can result in cardiac dysfunction. Our data showed that Tecrl deficiency results in decreased mitochondrial biogenesis and decreased Mfn2 protein (Figs. 4c, d, 5b, 6e, and 7b). We speculate that loss of mitochondrial homeostasis may disrupt MAM ultrastructure, which further exacerbates and disrupts calcium handling.

Sustained tachycardia and rapid pacing can lead to congestive heart failure and activate the anti-apoptotic PI3K cascade[33]. The PI3K/Akt pathway has been implicated in survival signaling in response to ischemia and reperfusion, oxidative stress, hypoxia, and β-adrenergic stimulation[33]. It was reported that drug everolimus could reverse aberrant $Akt^{ser473}$ and $P70S6$ signaling to slow mutant the proliferation of mutant fibroblasts and reverse mitochondrial abnormalities[34]. The Mohanraj group also showed that cannabidiol attenuated the generation of high glucose-induced reactive oxygen species, nuclear factor-κB activation, and cell death in primary human cardiomyocytes[35]. We hypothesize that mitochondrial dysfunction may cause apoptosis-inducing factor (AIF) and cytochrome C (Cyc) to be released from mitochondria into the cytoplasm, which then leads to apoptosis (Figs. 3c, 7). Interestingly, we noted that Akt signaling was inactivated in TECRL deficient cardiomyocytes (Figs. 5a, 6d). More importantly, TECRL overexpression induced mitochondrial respiration in a PI3K/Akt-dependent manner (Fig. 4c, d and Fig. S4e, f). Together, the PI3K/AKT-TECRL axis may function to regulate both mitochondrial respiration and cardiomyocyte survival.

In conclusion, we revealed that TECRL deficiency amplifies oxidative stress, results in loss of mitochondria ultrastructure and function, which is a hallmark of heart failure and aging[36]. Using Tecrl KO mice, hiPSC-CMs, and H9C2 cell models, we revealed that mitochondrial dysfunction is correlated with decreased MFN2, p-AKT (Ser473), and NRF2 protein levels. Overexpression of TECRL enhanced mitochondrial respiration in a PI3K/AKT-dependent manner. Our work provides support for targeting TECRL in treating CPVT.

**Limitations**. From the CPVT family clinical blood samples[6], we found that the TECRL protein level of the patient was much lower than his parents' (Fig. S1c), indicating that a Tecrl KO mouse model can mimic clinical phenotypes of CPVT disease. However, we must still construct the point mutation mouse model and further test them for arrhythmias.

## Methods

**Experimental procedures and animals**. WT and Tecrl KO mice were used for all experiments at the age of 4 to 5 and 8 weeks old. Animals were raised under controlled conditions (12 h dark-light cycle, 22 ± 2 °C, and 45–55% relative humidity) and housed with a maximum of five animals per cage, with unrestricted access to food and water. All animal studies were approved by the Institutional Review Board of Shanghai Children's Hospital, Shanghai Jiaotong University.

**The Tecrl KO mice were generated by CRISPR/Cas9**. The Tecrl KO mice were generated by Biocytogen (Beijing, China) using CRISPR/Cas9. Two single guide RNAs (sgRNAs) were designed to target the upstream of exon 3 and downstream of exon 4 of Tecrl using the CRISPR design tool (http://www.sanger.ac.uk/htgt/wge/). The sequences of sgRNAs 9 are as follows: Tecrl-T1, TTAGGATATGCCCTAG ATAA + AGG; Tecrl-T2, GATCGAAATACCTACGATCG + AGG. The in vitro-synthesized sgRNAs and Cas9 mRNA was co-injected into C57BL/6 mouse zygotes. After injection, surviving zygotes were then implanted to induce pregnancy. The genotype of Tecrl KO mice was confirmed by PCR amplification and DNA sequencing. Tail tip of each mouse was collected to isolate genomic DNA. The targeting region was amplified by PCR (primers: forward 5'-CATAGGGACTCG ATTGTTGTCCGTG-3', reverse 5'-TGCACTTGAATGGAAAAAGACTGGA-3'). Primer sequences of the WT and Tecrl KO mice for tail DNA identification are shown in Fig. S6a. And their positions are shown in Fig. S6b.

**Echocardiography**. Two-dimensional mouse echocardiography (Vevo 2100 ultrasound device, Visual Sonics Inc., Canada) was performed to test the left ventricular function, as previously described by Hu et al.[37] and Gao et al.[38]. Mice were anesthetized with isoflurane (1%) and moved to a workbench. A linear array probe and 40.0 MHz of center frequency were utilized. All measurements were averaged over five consecutive cardiac cycles and M-mode images of the left ventricle were recorded. LVIDs, LVIDd, LVESV, LVEDV, LVEF, and LVFS were measured to evaluate heart function.

**Surface electrocardiogram**. Animals were anesthetized by intraperitoneal injection of pentobarbital (100 mg/kg) and placed on a heating pad. Respiratory rate and loss of toe-press reflex were used to monitor the level of anesthesia. Non-invasive small animal ECG was performed in mice using equipment purchased from INDUS Technology (INDUS Technology, Inc, USA). After a stable baseline was reached, the ECG was recorded for 5 min. Epinephrine (2 mg/kg) and caffeine (120 mg/kg) were administered by intraperitoneal injection and ECG was continuously recorded in a time frame of 15 min. The entire post-epinephrine and caffeine ECG traces were used to analyze the presence or absence of ventricular arrhythmia. Information on the frequency and duration of arrhythmias was double-blind recorded and analyzed by skilled clinicians.

**RNA extraction and high-throughput sequencing**. RNAiso (Takara, Beijing, China) was utilized to extract total RNA from mouse ventricle tissues. RNA integrity was evaluated by the Bio-analyzer 2100 system (Agilent Technology, CA, USA). Ribosomal RNA was isolated from 3 μg of RNA using a commercially available RNA Removal Kit (Epicentre, WI, USA). Thereafter, the sequencing library was constructed. PCR products were purified and library qualification was detected. The library was sequenced using the Illumina Hiseq 3500 platform to generate 150 bps long paired-end reads. Raw and clean data were obtained after filtering for quality control. Reading counts for every sample were analyzed using HTSeq v6.0. RPKM (reads per kilobase million mapped reads) and computed to estimate gene expression levels. The GO and KEGG database (http://www.genome.jp/kegg/) were used to identify the aberrantly expressed genes. Benjamini-corrected $p < 0.05$ was used as the cut-off for significantly enriched biological processes.

**Isolation of adult mice cardiomyocytes and mitochondrial fractions**. Adult mice cardiomyocytes were isolated from the WT and Tecrl KO mouse ventricle (both right and left ventricles) tissues, following a previously published protocol[39,40]. Each mouse was injected with 150 or 300 units of heparin over 20 min. The mice were anesthetized by intraperitoneal injection of 100 mg/kg pentobarbital. The heart was quickly separated from the lungs and placed in an ice-cold buffer. The lungs were then removed. The ascending aorta was uncovered, then cannulated via the aorta. The aorta was tied and mounted on a modified Langendorff perfusion system. Subsequently, the heart was perfused with 37 °C oxygenated Krebs-Henseleit buffer solution minus calcium, and then perfused with 40 U/mL Collagenase Type II (Thermo Fisher, USA) and 50 mg/mL Collagenase Type IV (Sigma, German) for 30 min. Afterward, the ventricle was cut into small pieces for digestion under gentle agitation in the enzyme solution. Rod-shaped adult cardiomyocytes were collected by centrifugation, followed by the gradual addition of $CaCl_2$. Freshly isolated cardiomyocytes were plated on 25-mm coverslips pre-coated with 40 μg/mL laminin for 1 h (Thermo Fisher, USA). All cardiomyocytes were cultured in M199 medium (Sigma, USA) supplemented with 10% fetal bovine serum (Gibco, USA) at 37 °C and 5% $CO_2$ in a humidified incubator. Only excitable, rod-shaped, quiescent cells were used for the experiments. Mitochondria were isolated from heart tissues (both right and left ventricles) using a Mitochondria Isolation Kit (Applygen, China) according to the manufacturer's

instructions. Heart tissue was ground with a grinding rod and washed with phosphate-buffered saline (PBS) twice at room temperature. Homogenate and cells were pelleted by centrifugation at $1000 \times g$ for 15 min at 4 °C to break the cell membranes. Then the supernatant was recentrifuged three times to collect mitochondrial fractions.

**Electronic speculum assay**. TEM for morphological analysis was performed at Shanghai Institute Precision Medicine, Ninth People's Hospital, Shanghai Jiaotong University School of Medicine, according to standard operating procedures. For morphological TEM, heart tissues were cut into $1 \times 1 \times 1$ mm sized patches and fixed with ice-cold 2.5% glutaraldehyde at 4 °C overnight. Ultrathin sections were stained with uranyl acetate and lead citrate. After sample preparation, 90–100 nm thick sections were mounted on a 200-mesh copper grid and imprinted using an FEI Tecnai G2 Spirit transmission electron microscope. All analyses were performed blind to the observer.

**Seahorse assay**. Analyses of mitochondrial fuel usage were performed using Agilent Seahorse XF Cell Mito Stress Test Kits (Agilent Technologies, Santa Clara, CA, USA) according to the manufacturer's instructions. The day prior to the assay, H9C2 cells (5000/well)/hiPSC-CMs (4000/well) were seeded in Seahorse XFp96 cell culture miniplates 48 h before measurements (isolated mice cardiomyocytes (3000/well) were seeded 1 h before measurements). A sensor cartridge was hydrated in Seahorse XF Calibrant at 37 °C in a non-CO₂ incubator overnight. On the day of the assay, the XF Cell Mito Stress Test medium was prepared (XF Base Medium, 1 mM pyruvate, 2 mM glutamine, and 10 mM glucose warmed up at 37 °C and adjusted at pH 7.4 with 0.1 M NaOH). The above cells were transferred and incubated with the appropriate assay medium for 1 h in a non-CO₂ incubator at 37 °C. During the incubation time, the pouches containing the compounds (oligomycin, FCCP, trifluoromethoxy carbonyl cyanide phenylhydrazone, and rotenone/antimycin) were set at room temperature for 15 min. Compounds were resuspended with the prepared assay medium, then diluted in the same medium to obtain the following final concentrations: 10 μM oligomycin, 10 μM FCCP, and 5 μM rotenone/antimycin for the Mito Stress Test. The assay was conducted in a Seahorse XFp System and analyzed using Wave software (Agilent Technologies, Wilmington, USA).

**Primary neonatal cardiomyocytes culture**. The primary culture of neonatal cardiomyocytes was established according to the method described above[41] [10]. Experiments were approved by the Experimental Research Ethics Committee of Shanghai Children's Hospital affiliated to Shanghai Jiaotong University. Briefly, hearts from 1-day-old Sprague-Dawley rats were extracted and washed with cold PBS three times to remove blood. The hearts were sliced and then incubated multiple times in trypsin solution at 37 °C. After centrifugation, cell pellets were resuspended in Dulbecco's modified Eagle medium (Gibco, Grand Island, NY) containing 10% heat-inactivated fetal bovine serum (Gibco, Grand Island, NY). To increase the purity of cardiomyocytes, dissociated cells were plated in 100 mm culture dishes and stored in a 5% CO₂ incubator at 37 °C for 1 h. Nonmyocytes were readily attached to the bottom of the dish, while most cardiomyocytes remained suspended in the medium. The resulting suspension of cardiomyocytes was diluted and plated in confocal dishes for ROS detection. 5-Bromo2′-deoxyuridine (100 μM) was added during the first 48 h to prevent the proliferation of residual nonmyocytes.

**Fluorescent detection of ROS and Mitosox**. Intracellular ROS levels in the heart (both right and left ventricles) and neonatal cardiomyocytes were measured using DHE staining (Sigma-Aldrich, German). DHE powder was dissolved in dimethyl sulfoxide and diluted with PBS at 55 °C. Mouse heart tissue sections (7 μm) were obtained by using a frozen tissue slicer (Leica, German). Neonatal cardiomyocytes were incubated with 1 μM DHE for 30 min and observed under a laser confocal microscope (Zeiss LSM710, Germany) at a wavelength of 488/610 nm. For Mitosox assay, neonatal cardiomyocytes were incubated with 5 μM Mitosox for 30 min and analyzed using a PE PerkinElmer Operetta CLS at a maximum excitation/emission of approximately 510/580 nm.

**Mitochondrial extraction**. H9C2 cells were trypsinized and centrifuged for 5 min at 1000 rpm. H9C2 cells were washed and resuspended with ice-cold PBS, followed by counting and centrifugation for 5 min at $600 \times g$ at 4 °C. We then discarded the supernatant and added 1.5 mL extraction buffer to every $2$–$5 \times 10^7$ cells before incubating them in ice for 10–15 min. The supernatant was transferred to a fresh tube after homogenization, and centrifugation was carried out at $11,000 \times g$ for 10 min at 4 °C. The supernatant was carefully removed, for mitochondrial protein identification using a 100 μL cell lysis reagent and a cocktail of protease inhibitors (1:100 [V/V]) in the form of suspended particles.

**LC-MS/MS analyses and bioinformatics analysis**. Whole ventricular muscle mitochondria were lysed in ice for 30 min and centrifuged at 15000 rpm for 15 min at 4 °C. Then the supernatant was collected. The peptides were dissolved in a 5% acetonitrile aqueous solution containing 0.5% formic acid and analyzed by online

nanospray LC-MS/MS on Q Exactive™ (Thermo, USA) coupled to the Nano ACQUITY UPLC system (Waters Corporation, USA). The mass spectrometer was run in the data-dependent acquisition mode, and it automatically switched between MS and MS/MS mode. The data of the proteomics analysis were subjected to functional network and pathway enrichment analyses. Protein-protein network analysis and visualizations were performed using Cytoscape (3.7.2). Literature-curated pathway enrichment was achieved using QIAGEN's Ingenuity Pathway Analysis[42]. For GO-term pathway analysis, enrichment was reported as the percentage of total genes that annotate a given pathway (% enrichment), along with the Bonferroni Hochberg adjusted $p$ value.

**Cell lysate and immunoblotting**. To acquire cell proteins for immunoblotting analysis, hiPSC-CMs and H9C2 cells were washed twice in PBS and incubated with cell lysate buffer at 4 °C for 5 min while rocking gently. Proteins were extracted from left ventricular myocardial tissue and lysed with cell lysate buffer. The cell lysate buffer contained RIPA, protease inhibitors, and phosphatase inhibitors. Proteins were resolved by SDS-PAGE and transferred onto polyvinylidene fluoride membrane (Millipore, Bedford, MA, USA). The membranes were incubated with primary antibodies specific to Tecrl (Aviva Systems Biology, USA), Mfn1, Mfn2, Nrf2, Akt, p-Akt (Ser473), Fas, and GAPDH (Cell Signaling Technology, USA) in blocking buffer at dilutions of 1:1000. Membranes were incubated with secondary antibodies in a blocking buffer at 1:2000 for horseradish peroxidase (HRP)-conjugated antibodies (Cell Signaling Technology, USA). After washing, membranes were visualized using chemiluminescent substrate (ECL). Densities of immunology bands were analyzed using a scanning densitometer (GS-800, Bio-Rad Laboratories, Hercules, CA, USA) coupled with Bio-Rad personal computer analysis software.

**Human-induced pluripotent stem cell-based models**. A healthy volunteer provided informed consent to participate in this study, under protocols approved by the Shanghai Children's Hospital Institutional Review Board. Peripheral blood mononuclear cells were reprogrammed to pluripotency using the CytoTune Sendai reprogramming kit (Thermo Fisher, USA). These colonies were then stained for the pluripotency marker OCT4. HiPSCs were differentiated from hiPSC-CMs according to the manufacturer's protocol[15,18]. All the hiPSCs were maintained in Nutristem medium (Biological Industries, Israel) and passaged in Accutase enzyme cell detachment medium (Gibco, USA) every three to five days. Culture dishes were pre-coated with Matrigel (BD, USA) and diluted at 1:400. After at least 20 to 30 passages, hiPSCs were seeded to differentiation to hiPSC-CMs based on the timeline. On days 1–2, we added Chir (4–6 μM) to the aliquot of media 1. On days 3–4, we added IWR (5 μM) to the aliquot of media 1. On days 5–6, we replaced the media in each well with media 1 without added chemicals. On days 7–10, we replaced the media in each well with media 3. On days 11–13, we replaced the media in each well with media 2. During days 10–15, we checked for contraction under a microscope, as this should be visible if differentiation has been successful. Between day 14 and day 15, the cells were replaced onto a fresh plate to remove fibroblasts and other cell types that were generated during cardiomyocyte differentiation. A video of cardiomyocytes (hiPSC-CMs) spontaneously beating was captured using an Olympus inverted microscope (Olympus, Japan) and played with the Windows media player. MatLab and ImageJ, both open source softwares, were used to analyze the hiPSC-CMs beating video and evaluate the contractions. For a period of 2 s, the hiPSC-CMs beating video was recorded, and the contraction speed was determined using ImageJ Musclemotion-plugin, as described in a previously published method[43].

**Immunofluorescence analyses**. After permeabilization with triton-X, the hiPSCs or hiPSC-CMs were stained with OCT4 antibody (Cell Signaling Technology, USA), or α-actinin antibody (Cell Signaling Technology, USA). Images were acquired with a laser confocal microscope (Zeiss LSM710, German).

**Statistical analysis**. Tandem mass spectra were processed in PEAKS Studio version X (Bioinformatics Solutions Inc, Canada). Peptides were filtered using 1% FDR and one unique peptide. Differently expressed proteins were filtered if they contained at least one unique peptide with significance over 13 ($p < 0.05$), and if their fold change was over 1.3. The results were expressed as mean ± SEM. Statistical analysis was performed using SPSS software, version 21.0 (SPSS, Inc., USA). Comparisons among groups were performed by one-way ANOVA. Paired data were evaluated using two-tailed Student's $t$-tests. Statistical significance was set at $p < 0.05$.

**Reporting summary**. Further information on research design is available in the Nature Research Reporting Summary linked to this article.

## Data availability

All relevant data are available by request from the authors. RNA-seq, cardiac mitochondrial proteome, and TECRL overexpression mass spectrometry data have been deposited in Figshare.com and can be accessed with the accession code 14375606.v1. The supplementary movie (Video S1) was deposited on Figshare.com and can be accessed (https://figshare.com/s/42294262dafe725b2f45). Supplementary tables (Tables S1–S6) were deposited in Figshare.com and can be accessed (https://figshare.com/s/

7c549032e208a67c1fa7). Table S1. Mass spectrometry data. Table S2. RNA-sequencing data. Table S3. Mitochondrial proteomics data. Table S4. HiPSC-CMs contraction data. Table S5. Mass spectrometry of hiPSC-CMs data. Table S6. Mass spectrometry of co-immunoprecipitation overexpressed by TECRL in hiPSC-CMs. Supplementary figures and all the uncropped blots (Fig. S7) were in the supplementary information file.

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

## Acknowledgements

We thank the imaging platform and electron microscope platform staff of Shanghai Institute Precision Medicine, Ninth People's Hospital, Shanghai Jiaotong University School of Medicine. This work was supported by the National Natural Science Foundation of China (NSFC) (No. 81900437), the Shanghai Jiaotong University medical technology crossing project (YG2021ZD26 and ZH2018ZDA26), the Shanghai Science and Technology Committee (18411965800 and 19411963600), and Shanghai Children's Hospital (2019YN006). No benefit in any form has been or will be received from a commercial organization directly or indirectly.

## Author contributions

T.X., C.H., and A.C.Y.C. designed and operated the project. X.J., M.X., and Y.Z. provided the clinical Echocardiography analysis. C.H. wrote the manuscript with input from L.X., X.J., H.Z., X.S., J.Z., and Q.Q. L.X. designed and supplemented the related experiments in the revised manuscript. All authors read and approved the final manuscript.

## Competing interests

The authors declare no competing interests.
