## [Peer Review File · Communications Biology]

Reviewers' comments:

Reviewer #1 (Remarks to the Author):

Authors developed KO mice for TECRL. Mutation within the protein was previously associated with Catecholaminergic polymorphic VT. They demonstrated that mice are susceptible to structural heart remodeling. It was associated with MFN2, p-AKT (Ser473), and NRF2., increased ROS production and reduced mitochondria performance. Using IPCS they demonstrate that TECRL regulates mitochondria function.

There is novelty in the manuscript, but improvement is necessary in order to fulfill the claims asked by the authors. Please, find below my comments.

The first claim in the discussion section is " Here we provide characterization of a novel CPVT Tecrl murine model.". Also, the conclusion of the manuscript is "Our work provides a novel CPVT murine model and provides support for targeting TECRL in treating CPVT." Catecholaminergic polymorphic VT (CPVT). CPVT is a rare arrhythmogenic disorder characterized by adrenergic-induced bidirectional and polymorphic VT. If authors claim that they developed a CPVT model, they have to demonstrate that mice develop the phenomena. There is no electrophysiological experiment to support the claim in the manuscript. Furthermore, it is necessary because as demonstrated by Figure 1E there is still some expression of TECRL. Previous studies have demonstrated that patients with heterozygosity for TECRL does not develop CPVT (1, 2) Also, implication of partial KO of the protein has to be discussed as a limitation in the study involving mice.

(1) TECRL, a new life-threatening inherited arrhythmia gene associated with overlapping clinical features of both LQTS and CPVT. *EMBO Mol Med*. 2016 Dec 1;8(12):1390-1408.

(2) Life-threatening arrhythmias with autosomal recessive TECRL variants. *Europace*. 2020 Dec 25;euaa376. doi: 10.1093/europace/euaa376.

Human induced pluripotent stem cell-based models

Authors provided only one molecular marker to affirm that they obtained cardiomyocytes from induced pluripotent stem cells. Functional analysis should be provided. Please, provide as supplementary material a movie showing that your cells are spontaneously contracting at the stage you are analysing them. Also, provide an analysis showing how your molecular manipulations impact the contraction of CM-hiPSCs. You may use the free software below to analyse your movies. (3)

(3) MUSCLEMOTIONA Versatile Open Software Tool to Quantify Cardiomyocyte and Cardiac Muscle Contraction In Vitro and In Vivo. *Circ Res*. 2018 Feb 2; 122(3): e5–e16.

In the introduction section authors inform that the pathogenic mechanism remains to be elucidated in TECRL mutation induce CPVT. However, the manuscript (1) already showed a good description of the molecular pathogenesis in this disease. Please, rewrite your phrase informing that additional mechanisms may play a role.

Line 70 "elevated reactive oxygen species". What this mean?

Authors claim that mitochondrial fusion was inhibited in Tecrl-deficient cardiomyocytes. However, there is non histology experiment to support it. Please, change your claim.

A probe of 10 MHz to perform Ecocardiography was used. For mice it is indicated that a higher frequency probe should be used. Please, discuss this limitation in your manuscript <https://www.ncbi.nlm.nih.gov/pmc/articles/PMC3130310/>

Isolation of adult mice cardiomyocytes and mitochondrial fractions

Please, indicate the region of the heart that you obtained your isolated cardiomyocytes.

Fluorescent detection of ROS

Please, provide a better description about your analysis. How did you identify the cardiomyocytes to ensure that enhanced ROS level is attributed to cardiomyocytes? As your result is presented, enhanced ROS level could be attributed to other cell types in the heart tissue. Analysis of ROS level in isolated cardiomyocytes has to be performed. Also, your figures do not show an enhanced level of ROS.

LC-MS/MS analyses and bioinformatics analysis

Please, identify if you analysed right, left, septum or the whole ventricular tissue.

Cell lysate and immunoblotting

Please, specify in the method the cell type used for this analysis.

Reviewer #2 (Remarks to the Author):

The present manuscript claims that TECRL regulates mitochondrial function mainly through PI3K/AKT signaling and mitochondrial fusion, and that its deficiency is at the basis of CPVT disorder. For this, the authors assessed changes in the expression of various proteins related with several pathways, ROS production, echocardiography parameters, and mitochondrial homeostasis. I have some major concerns:

- 1) I would like the authors to describe the changes in apoptosis-related proteins observed in the KEGG pathway analysis (. Discuss if changes in bcl2 family proteins are the egg or the chicken associated with the mitochondrial dysfunction.
- 2) Fig. 1E shows a band of TECRL in TECRL KO mice (almost 30% respect to WT mice). Why is this? It seems to be that the protein is knocked-down instead of knocked-out. Please discuss.
- 3) Which is the number of experiments in figure S2B ? Because the means seems to be very similar to those in Fig. 4F, but with higher error. Is it a sample size issue?
- 4) Which is the mechanism by which the reduction in TECRL expression changes the protein levels of MFN2, phosphorylation state of AKT, and NRF2 (hiPSC-CM), in addition to changes in genes involved in rhythmic process, regulation of circadian rhythm, circadian rhythm, glycogen biosynthetic process, and glucose metabolic process (TECRL KO mice)? The present manuscript is descriptive but lacks of solid mechanistic description by which TECRL changes many of the described protein levels.
- 5) Do the authors have any experiments in which TELCR is silenced instead of constitutively knocked-out/down? What would they speculate to happen in a model of iRNA-mediated silencing of TECRL?
- 6) Fig. 7D: Which is the statistical significance of the difference between "Ctrl maximal respiration" and "Ctrl + LY294002 maximal respiration"? If it is statistically significant, how do you explain that increase in max resp when inhibiting PI3K/Akt?
- 7) The authors refer to a novel murine model of CPVT, but previous work of them refer to as a model of LQTS/CPVT. In addition, in line 235 the authors say "Our work provides a novel CPVT murine model...". Although they have previous work describing the TECRL-related LQTS/CPVT model, in the present manuscript they do not present any experiment showing related to arrhythmias. Please rephrase the sentence.

Minor concerns:

Please correct the following:

Line 126: assayed

Line 140: it should say: ...arrhythmogenic right ventricular "cardiomyopathy"...

Lines 63-64: Please revise sentence. It has no sense.

Lines 145-147: Please revise first sentence. It has no sense.

Line 38: please delete space in number. Collection refers to review? Collaboration?...

Line 179: correct "mitochondrial"

Line 213: correct "Mitochondiral"

Line 192-195: are the authors referring to mitochondrial dysfunction? Please revise the whole sentence.

Lines 184-185: Please revise sentence.

Line 157: α -actinin instead of α -actinnin

There are several sentences/asseverations that need a revision by a native English speaker.

Review report and point to point replies:

Reviewer #1 (Remarks to the Author):

Authors developed KO mice for TECRL. Mutation within the protein was previously associated with Catecholaminergic polymorphic VT. They demonstrated that mice are susceptible to structural heart remodeling. It was associated with MFN2, p-AKT (Ser473), and NRF2., increased ROS production and reduced mitochondria performance. Using IPCS they demonstrate that TECRL regulates mitochondria function.

There is novelty in the manuscript, but improvement is necessary in order to fulfill the claims asked by the authors. Please, find below my comments.

The first claim in the discussion section is " Here we provide characterization of a novel CPVT Tecrl murine model.". Also, the conclusion of the manuscript is "Our work provides a novel CPVT murine model and provides support for targeting TECRL in treating CPVT. "Catecholaminergic polymorphic VT (CPVT). CPVT is a rare arrhythmogenic disorder characterized by adrenergic-induced bidirectional and polymorphic VT. If authors claim that they developed a CPVT model, they have to demonstrate that mice develop the phenomena. There is no electrophysiological experiment to support the claim in the manuscript. Furthermore, it is necessary because as demonstrated by Figure 1E there is still some expression of TECRL. Previous studies have demonstrated that patients with heterozygosis for TECRL does not develop CPVT (1, 2) Also, implication of partial KO of the protein has to be discussed as a limitation in the study involving mice.

(1) TECRL, a new life-threatening inherited arrhythmia gene associated with overlapping clinical features of both LQTS and CPVT. *EMBO Mol Med.* 2016 Dec 1;8(12):1390-1408.

(2) Life-threatening arrhythmias with autosomal recessive TECRL variants. *Europace.* 2020 Dec 25;euaa376. doi: 10.1093/europace/euaa376.

Response: Thanks for your advice. As for the question: The first claim in the

discussion section is "Here we provide characterization of a novel CPVT Tecrl murine model". Also, the conclusion of the manuscript is "Our work provides a novel CPVT murine model and provides support for targeting TECRL in treating CPVT." We are sorry for the poor description. We have rewritten in the reversed manuscript. See line 216-219, page 10. Line 287-287, page 12.

For the question "Figure 1E there is still some expression of TECRL". Since 2017, we have constructed this Tecrl KO mouse. At that time, there was only one TECRL antibody (Catalog No: ARP44592_P050, Aviva systems biology). Based on the mass spectrometry of the WT and Tecrl KO mice heart tissues (Table S1), TECL and TECRL were found in the WT mice, only TECR was found in Tecrl KO mice. As TECR and TECRL both belongs to the steroid 5-alpha reductase family, and western blotting is a semi- quantitative experiment, we still detected a little TECRL expression. Based on our results, this is a Tecrl KO mouse model. And this is one of the shortages of the manuscript. We have added to the shortage in the reversion manuscript. See line 93-96, page 5. Line 290-296, page 12.

For the question "demonstrate that mice develop the CPVT phenomena". We have performed the mice surface electrocardiogram (ECG) in another manuscript (CVR-2021-1489, Cuilan Hou, and Tingting Xiao) in Cardiovascular Research. Here we just exhibited the representative surface ECG pictures. In the Tecrl KO mice, multiple premature ventricular beats and ventricular tachycardia (VT) were observed immediately after an epinephrine and caffeine (epi/caffeine) injection; however, few ventricular events were observed in the WT mice (Fig. 2C-D), indicating that the TECRL KO mice exhibited cardiac arrhythmias, which can mimic CPVT phenotypes. See line 104-107, page 5.

Surface electrocardiogram

Animals were anesthetized by intraperitoneal injection of pentobarbital (100 mg/kg) and placed on a heating pad. Respiratory rate and loss of toe-press reflex were used to monitor the level of anesthesia. Non-invasive small animal ECG was performed in mice at 4 and 8 weeks of age using equipment purchased from INDUS Technology (INDUS Technology, Inc, USA). The baseline ECG was recorded for 5 min, followed

by an additional 15 min following an intraperitoneal injection of epinephrine (2 mg/kg) and caffeine (120 mg/kg). See line 338-345, page 14.

In the *Tecr1* KO mice, multiple premature ventricular beats and ventricular tachycardia (VT) were observed immediately after an epinephrine and caffeine (epi/caffeine) injection; however, few ventricular events were observed in the WT mice (Fig. 2C-D).

See line 104-107, page 5.

Fig 2. *Tecr1* deficiency induced cardiac dysfunction. (A) Representative echocardiography images of the WT and *Tecr1* KO mice. (B) Quantification of LVEF, LVFS, LVIDd, LVIDs, LVEDV, and LVESV in the WT and *Tecr1* KO mice (8 weeks) (n = 8). C-D. Representative ECG recordings in the WT and *Tecr1* KO mice before and after epi/caffeine stimulation (8 weeks) (n = 6). Values are mean \pm SE. P < 0.05

was considered significant. See line 567-568, page 22. Line 802-803, page 33.

As for the question “previous studies have demonstrated that patients with heterozygosis for TECRL does not develop CPVT”, Devalla *et al* reported that genomic DNA from the parents of IV:2 (III:1 and III:2; Fig 1D), who are clinically normal, was also included for whole-exome sequencing ^[1]. In the above report, we found that the parent was just carried only one mutation of the TECRL gene (c.331+1G>A), or TECRL gene (p. Arg196Gln), in our study, the patient father was normal, who just carried TECRL gene (c.918+3T>G), and his mother also normal, who just carried TECRL (p. Arg196Gln), however, the patient had CPVT clinical symptoms and had a compound heterozygosity of TECRL gene (p. Arg196Gln, c.918+3T>G). In our previous study ^[2], based on support vector machine and neural network analysis predicted that Arg196Gln mutation could decrease the stability of Tecrl structure. A STRUM server also confirmed that Arg196Gln mutation may decrease the binding capacity of the substrate and cause an amino acid substitution immediately upstream of the 3-oxo-5-alpha steroid 4-dehydrogenase domain. According to the “human splicing finder” indication and Alamut Visual Splicing Prediction, the c.918 + 3T>G mutation could influence TECRL variable splicing. Based on these prediction results and the patient clinical symptom, the TECRL compound heterozygosity may be associated with CPVT. If the patient and his parents peripheral blood mononuclear cells can be reprogrammed into cardiomyocytes and then detected their calcium transients, after a series of experiments, then we concluded that this complex heterozygous mutation may be related to CPVT.

References:

- [1] Devalla HD, Gelinas R, Aburawi EH, Beqqali A, Goyette P, Freund C, *et al.*. TECRL, a new life-threatening inherited arrhythmia gene associated with overlapping clinical features of both LQTS and CPVT. EMBO MOL MED 2016;8:1390-1408.
- [2] Xie L, Hou C, Jiang X, Zhao J, Li Y, Xiao T. A compound heterozygosity of Tecrl gene confirmed in a catecholaminergic polymorphic ventricular tachycardia family. EUR J MED GENET 2019; 62:103631.

Human induced pluripotent stem cell-based models

Authors provided only one molecular marker to affirm that they obtained cardiomyocytes from induced pluripotent stem cells. Functional analysis should be provided. Please, provide as supplementary material a movie showing that your cells are spontaneously contracting at the stage you are analysing them. Also, provide an analysis showing how your molecular manipulations impact the contraction of CM-hiPSCs. You may use the free software below to analyse your movies. (3)

(3) MUSCLEMOTIONA Versatile Open Software Tool to Quantify Cardiomyocyte and Cardiac Muscle Contraction In Vitro and In Vivo. *Circ Res.* 2018 Feb 2; 122(3): e5–e16.

Response: Agree. We added another molecular marker (cTnT) to confirm the cardiomyocytes from induced pluripotent stem cells in the revised Fig. 6B. We have added in the reversion manuscript.

A movie showing that the cardiomyocytes (hiPSC-CMs) are spontaneously contracting was added in the supporting information in the revised manuscript. See line 178-185, page 8. Line 599, page 23. Line 816-817, page 38-39.

As for the question “Functional analysis should be provided”, we added the following experiments.

A video of cardiomyocytes (hiPSC-CMs) spontaneously beating was captured using an Olympus inverted microscope (Olympus, Japan) and played with the Windows media player. MatLab and Image J, both open source softwares, were used to analyze the hiPSC-CMs beating video and evaluate the contractions. For a period of 2 s, the hiPSC-CMs beating video was recorded, and the contraction speed was determined using Image J Musclemotion-plugin, as described in a previously published method⁴³. See line 509-515, page 19-20.

Results: We measured the baseline of contraction of hiPSC-CMs. Statistical analysis showed no significant difference between the Cth and shTECRL group (Table S4). Musclemotion was used to determine the speed of contraction of the hiPSC-CMs beating in a two-second-long video. The contraction amplitude of knock-down hiPSC-CMs was higher compared to the control hiPSC-CMs, but there was no

significance between the Cth and shTECRL group (Fig. 6C). See line 179-185, page 8.

In the introduction section authors inform that the pathogenic mechanism remains to be elucidated in TECRL mutation induce CPVT. However, the manuscript (1) already showed a good description of the molecular pathogenesis in this disease. Please, rewrite your phrase informing that additional mechanisms may play a role.

Response: Agree. Thanks for your advice. We have rewritten in the reversion manuscript. See line 54-55, page 3. Line 70-71, page 3.

Line 70 "elevated reactive oxygen species". What this mean?

Response: Agree. Reactive oxygen species (ROS) include superoxide and hydroxyl radicals, nitric oxide, singlet oxygen, nitrogen dioxide, and peroxynitrite. Major sources of ROS include NADPH oxidases, cyclooxygenases, and mitochondria.

Authors claim that mitochondrial fusion was inhibited in Tecrl-deficient cardiomyocytes. However, there is non histology experiment to support it. Please, change your claim.

Response: Agree. We have rewritten in the reversion manuscript. See line 28-29, page 2. Line 79, page 4. Line 163-164, page 7. Line 592, page 23.

A probe of 10 MHz to perform Ecocardiography was used. For mice it is indicated that a higher frequency probe should be used. Please, discuss this limitation in your manuscript

<https://www.ncbi.nlm.nih.gov/pmc/articles/PMC3130310/>

Response: Agree. We are sorry for the mistake. We have rewritten in the reversion manuscript. See line 331-332, page 13-14.

Isolation of adult mice cardiomyocytes and mitochondrial fractions

Please, indicate the region of the heart that you obtained your isolated

cardiomyocytes.

Response: Agree. We have added the region of the heart that we obtained the isolated cardiomyocytes. We have rewritten in the reversion manuscript. See line 364-365, page 15. Line 373, page 15.

Fluorescent detection of ROS

Please, provide a better description about your analysis. How did you identify the cardiomyocytes to ensure that enhanced ROS level is attributed to cardiomyocytes? As your result is presented, enhanced ROS level could be attributed to other cell types in the heart tissue. Analysis of ROS level in isolated cardiomyocytes has to be performed. Also, your figures do not show an enhanced level of ROS.

Response: Agree. Thanks for your advice. We have analyzed ROS levels in isolated primary neonatal cardiomyocytes, and we also detected Mitosox levels in isolated primary neonatal cardiomyocytes. We also found that DHE and Mitosox levels were increased upon TECRL knockout in primary neonatal cardiomyocytes (Fig. 4G-J). We have rewritten in the reversion manuscript. See line 143-144, page 6. line 417-431, page 16-17. Line 585-588, page 23. Line 809-810, page 35-36.

LC-MS/MS analyses and bioinformatics analysis

Please, identify if you analysed right, left, septum or the whole ventricular tissue.

Response: Agree. We are sorry for the unclear description. We have rewritten in the reversion manuscript. See line 458, page 18.

Cell lysate and immunoblotting

Please, specify in the method the cell type used for this analysis.

Response: Agree. We are sorry for the unclear description. We have rewritten in the reversion manuscript. See line 473-476, page 18.

Reviewer #2 (Remarks to the Author):

The present manuscript claims that TECRL regulates mitochondrial function mainly

through PI3K/AKT signaling and mitochondrial fusion, and that its deficiency is at the basis of CPVT disorder. For this, the authors assessed changes in the expression of various proteins related with several pathways, ROS production, echocardiography parameters, and mitochondrial homeostasis.

I have some major concerns:

1) I would like the authors to describe the changes in apoptosis-related proteins observed in the KEGG pathway analysis (Discuss if changes in bcl2 family proteins are the egg or the chicken associated with the mitochondrial dysfunction).

Response: Agree. Thanks for your advice. We have rewritten in the reversed manuscript. We think that mitochondrial dysfunction may cause apoptosis inducing factor (AIF) and cytochrome C (Cyc) released from the mitochondria into the cytoplasm, then it leads to apoptosis. See line 204-211, page 8-9. See line 444-453, page 17-18. See line 608-614, page 23-24. Line 820-821, page 40-41.

Methods

Mitochondrial extraction

H9C2 cells were trypsinized and centrifuged for 5 min at 1000 rpm. H9C2 cells were washed and resuspended with ice cold PBS, followed by counting and centrifugation for 5 min at $600 \times g$ at $4^\circ C$. We then discarded the supernatant and added 1.5 mL extraction buffer to every $2-5 \times 10^7$ cells before incubating them in ice for 10-15 min. The supernatant was transferred to a fresh tube after homogenization, and centrifugation was carried out at $11,000 \times g$ for 10 min at $4^\circ C$. The supernatant was carefully removed, for mitochondrial protein identification using 100 μ L cell lysis reagent and a cocktail of protease inhibitors (1:100 [V/V]) in the form of suspended particles.

Results

Based on the mass spectrometry of TECRL co-immunoprecipitation (Table. S6), we observed that ATP synthase subunit beta, mitochondrial (ATPB) was connected with TECRL, and the contraction was significantly increased after TECRL overexpression (Fig. 7C).

hiPSC-CMs and H9C2 cells were utilized to measure mitochondrial respiration.

Maximum mitochondrial respiration was significantly enhanced with TECRL overexpression in hiPSC-CMs, but was blocked upon PI3K/Akt inhibitor LY294002 treatment (Fig. 7D–E). Similarly, mitochondrial respiration in H9C2 cells was significantly enhanced (basal respiration, ATP production, and maximal respiration) upon TECRL overexpression, and this increase was also PI3K/Akt dependent (Fig. S4 E-F). Meanwhile, based on the cardiac RNA-seq, we observed that apoptosis-related proteins observed in the KEGG pathway analysis (Fig. 3C), we also found that apoptosis inducing factor (AIF) and cytochrome C (Cyc) were released from mitochondria into the cytoplasm after 60 nM siTECRL infection followed by immunofluorescent staining (Fig. 7F) and western blotting (Fig 7. G) in H9C2 cells. The 60 nM siTECRL infection increased the expression levels of Cyc and AIF in cytoplasm. There was no significance in mitochondrial Cyc and AIF levels in between 60 nM siTECRL infection group and its negative control (Fig 7. G).

2) Fig. 1E shows a band of TECRL in TECRL KO mice (almost 30% respect to WT mice). Why is this? It seems to be that the protein is knocked-down instead of knocked-out. Please discuss.

Response: Agree. We are sorry for the mistake.

Since 2017, we have constructed this Tecrl KO mouse. At that time, there was only one TECRL antibody (Catalog No: ARP44592_P050, Aviva systems biology). Based on mass spectrometry of the wild type (WT) and TECRL KO mice heart tissues (Table S1), we found that TECL and TECRL were present in WT mice, but only TECL was found in TECRL KO mice, indicating that the Tecrl KO mice had been produced successfully. As TECL and TECRL both belongs to the steroid 5-alpha reductase family and considering that western blotting is a semi- quantitative method, we still detected a low level of TECRL expression. Based on our results, this is a Tecrl KO mouse model. And this is one of the shortages of the manuscript. We have added to the shortage in the reversion manuscript. See line 93-96, page 5. Line 289-295, page 12.

3) Which is the number of experiments in figure S2B? Because the means seems to be very similar to those in Fig. 4F, but with higher error. Is it a sample size issue?

Response: Agree. There are eight samples from four mice of each group in Figure S2B. All of the eight samples contain more than 6 images, which were randomly photographed under a laser confocal microscope (Zeiss LSM710, Germany).

Then we added two mice in each group and re-analyzed. We have rewritten in the reversion manuscript. See line 827-828, page 43.

4) Which is the mechanism by which the reduction in TECRL expression changes the protein levels of MFN2, phosphorylation state of AKT, and NRF2 (hiPSC-CM), in addition to changes in genes involved in rhythmic process, regulation of circadian rhythm, circadian rhythm, glycogen biosynthetic process, and glucose metabolic process (TECRL KO mice)? The present manuscript is descriptive but lacks of solid mechanistic description by which TECRL changes many of the described protein levels.

Response: Agree. We have rewritten in the reversed manuscript.

Firstly, we did a lot of experiments, and we think that TECRL deficiency may cause mitochondria dysfunction, then results in oxidative stress (Fig. 4 and Fig. S2). Then NRF2 antioxidant stress system was disturbed after chronic oxidative stress.

Using shRNA knockdown, we observed loss of TECRL resulted in a decrease in protein levels of MFN2, p-AKT (Ser473), and NRF2 (Fig. 6D-G). We also confirmed that the mRNA (Fig. S4 A-D) and protein (Fig. S5 A-C) levels of MFN2 and NRF2 were decreased, via FAS increased in H9C2 cells. Next, we overexpressed TECRL in hiPSC-CMs, and mass spectrometry was performed. The mass spectrometry results showed FAS decreased significantly upon TECRL overexpression (Table S5). Results of immunoblotting confirmed that FAS protein decreased following TECRL overexpression (Fig. 7A), in agreement with our animal findings (Fig. 5D). Moreover, MFN2 was partially increased following TECRL overexpression (Fig. 7B). Based on the mass spectrometry of TECRL co-immunoprecipitation (Table. S6), we observed that ATP synthase subunit beta, mitochondrial (ATPB) was contacted with TECRL,

and the contact was much more after TECRL overexpression (Fig. 7C).

Secondly, Mohanraj group reported that diabetic cardiomyopathy was characterized by declined diastolic and systolic myocardial performance associated with increased oxidative-nitrative stress, nuclear factor- κ B and mitogen-activated protein kinase (c-Jun N-terminal kinase, p-38, p38 α) activation, enhanced expression of adhesion molecules (intercellular adhesion molecule-1, vascular cell adhesion molecule-1), tumor necrosis factor- α , markers of fibrosis (transforming growth factor- β , connective tissue growth factor, fibronectin, collagen-1, matrix metalloproteinase-2 and -9), enhanced cell death (caspase 3/7 and poly[adenosine diphosphate-ribose] polymerase activity, chromatin fragmentation, and terminal deoxynucleotidyl transferase dUTP nick end labeling), and diminished Akt phosphorylation¹. Mohanraj group reported that cannabidiol attenuated myocardial dysfunction, cardiac fibrosis, oxidative/nitrative stress, inflammation, cell death, and interrelated signaling pathways¹. Mohanraj group also showed that cannabidiol attenuated the high glucose-induced increased reactive oxygen species generation, nuclear factor- κ B activation, and cell death in primary human cardiomyocytes^[1].

Santull *et al.* showed that 27 RYR2 CPVT patients displayed abnormal oral glucose tolerance and attributed this observation to sarcoplasmic reticulum stress induced b-cell apoptosis^[2]. Bioinformatics analysis showed that the function of TECRL protein focused on very long-chain fatty acid biosynthetic processes and oxidoreductase activity. Here we show that loss of Tecrl results in altered Nrf2 and Fas expression changes (Fig. 3, S3 and 5C-D). Given that adult cardiomyocytes utilizes free fatty acids as the main metabolic substrate, mitochondrial dysfunction in TECRL deficient cardiomyocytes supports the notion of an alternative pathway driving CPVT^[3].

Thus, we think that TECRL deficiency may cause mitochondria dysfunction, then cause diminished Akt phosphorylation, the TECRL deficiency results cardiac pathological change, which similar to diabetic cardiomyopathy. We hypothesize that long-term TECRL deficiency leads to cardiomyopathy.

References:

[1] Rajesh M, Mukhopadhyay P, Batkai S, Patel V, Saito K, Matsumoto S, et al.. Cannabidiol attenuates cardiac dysfunction, oxidative stress, fibrosis, and inflammatory and cell death signaling pathways in diabetic cardiomyopathy. *J AM COLL CARDIOL* 2010;56:2115-2125.

[2] Santulli, G. et al. Calcium release channel RyR2 regulates insulin release and glucose homeostasis. *J. Clin. Invest.* **125**, 1968-1978 (2015).

[3] Bonnet, D. et al. Arrhythmias and conduction defects as presenting symptoms of fatty acid oxidation disorders in children. *Circulation.* **100**, 2248-2253 (1999).

5) Do the authors have any experiments in which TELCR is silenced instead of constitutively knocked-out/down? What would they speculate to happen in a model of iRNA-mediated silencing of TECRL?

Response: Agree. We have added the TECRL silenced related experiments. We have rewritten in the reversion manuscript. See line 186-188, page 8. Line 640-643, page 25. Line 834-837, page 45-46.

As siRNA did not have effects in primary neonatal cardiomyocytes and hiPSC-CMs, we use H9C2 cells to do following experiments. We designed and tested siTECRL infection efficiency, then we also confirmed that the mRNA (Fig. S4 A-D) and protein (Fig. S5 A-C) levels of MFN2 and NRF2 were decreased, via FAS increased in H9C2 cells.

6) Fig. 7D: Which is the statistical significance of the difference between “Ctrl maximal respiration” and “Ctrl + LY294002 maximal respiration”? If it is statistically significant, how do you explain that increase in max resp when inhibiting PI3K/Akt?

Response: Agree. We are sorry for the unclear statistical description. We added another experiment and combined the data with the previous ones. Then we found that there was no significance between Cth maximal respiration and Cth+LY294002, while there was statistical significance between Cth maximal respiration and OeTECRL. There was statistical significance between OeTECRL maximal respiration and OeTECRL+LY294002. Then the seahorse assay showed that mitochondrial maximal respiration was significantly enhanced upon TECRL overexpression in hiPSC-CMs,

but was blocked upon PI3K/Akt inhibitor LY294002 treatment (Fig. 7C–D). And the same tendency was also demonstrated in H9C2. We have reorganized the Figure 7 in the reversion manuscript. See line 820-821, page 40-41.

7) The authors refer to a novel murine model of CPVT, but previous work of them refer to as a model of LQTS/CPVT. In addition, in line 235 the authors say “Our work provides a novel CPVT murine model...”. Although they have previous work describing the TECRL-related LQTS/CPVT model, in the present manuscript they do not present any experiment showing related to arrhythmias. Please rephrase the sentence.

Response: Agree. We are sorry for the mistake.

We have performed the mice surface electrocardiogram (ECG) in another manuscript (CVR-2021-1489, Cuilan Hou, and Tingting Xiao) in Cardiovascular Research. Here we just exhibited the representative surface ECG pictures. In the Tecrl KO mice, multiple premature ventricular beats and ventricular tachycardia (VT) were observed immediately after an epinephrine and caffeine (epi/caffeine) injection; however, few ventricular events were observed in the WT mice (Fig. 2C-D), indicating that the TECRL KO mice exhibited cardiac arrhythmias, which can mimic CPVT phenotypes. See line 104-107, page 5.

Surface electrocardiogram

Animals were anesthetized by intraperitoneal injection of pentobarbital (100 mg/kg) and placed on a heating pad. Respiratory rate and loss of toe-press reflex were used to monitor the level of anesthesia. Non-invasive small animal ECG was performed in mice at 4 and 8 weeks of age using equipment purchased from INDUS Technology (INDUS Technology, Inc, USA). The baseline ECG was recorded for 5 min, followed by an additional 15 min following an intraperitoneal injection of epinephrine (2 mg/kg) and caffeine (120 mg/kg). See line 338-345, page 14.

In the Tecrl KO mice, multiple premature ventricular beats and ventricular tachycardia (VT) were observed immediately after an epinephrine and caffeine (epi/caffeine) injection; however, few ventricular events were observed in the WT

mice (Fig. 2C-D).

See line 104-107, page 5.

Fig 2. *Tecl1* deficiency induced cardiac dysfunction. (A) Representative echocardiography images of the WT and *Tecl1* KO mice. (B) Quantification of LVEF, LVFS, LVIDd, LVIDs, LVEDV, and LVESV in the WT and *Tecl1* KO mice (8 weeks) ($n = 8$). C-D. Representative ECG recordings in the WT and *Tecl1* KO mice before and after epi/caffeine stimulation (8 weeks) ($n = 6$). Values are mean \pm SE. $P < 0.05$ was considered significant. See line 567-568, page 22. Line 802-803, page 33.

Minor concerns:

Please correct the following:

Line 126: assayed

Line 140: it should say: ...arrhythmogenic right ventricular “cardiomyopathy”...

Lines 63-64: Please revise sentence. It has no sense.

Lines 145-147: Please revise first sentence. It has no sense.

Line 38: please delete space in number. Collection refers to review? Collaboration?...

Line 179: correct “mitochondrial”

Line 213: correct “Mitochondiral”

Line 192-195: are the authors referring to mitochondrial dysfunction? Please revise the whole sentence.

Lines 184-185: Please revise sentence.

Line 157: α -actinin instead of α -actinnin

Response: Agree. We are sorry for the mistake. We have rewritten in the reversion manuscript. See line 146, page 6; line 160, page 7; line69-70, page 4; 165-167, page 7; line 42, page 3; line 219, page 10; line 258, page 11; line 237-239, page 10; line 226-228, page 10; line 178, page 8.

There are several sentences/asseverations that need a revision by a native English speaker.

Response: Agree. The language of this manuscript has been improved through two steps. First, the authors read the manuscript again and corrected the writing. Then it was sent to Elsevier Language Editing Services for further language polishing with American English Style. Please see the revised manuscript and attached certificate.

Reviewers' comments:

Reviewer #1 (Remarks to the Author):

Dear authors

I appreciate your efforts to address my comments. However, I still have some comments on your manuscript.

From Supplementary Table 1 it is really difficult to understand the result.

Most importantly, it is not possible, as it is shown, to conclude that your KO mouse is indeed KO.

Please, better describe your result.

Also, at this stage I'm not 100% sure that your mice are indeed KO.

Since your antibody is not 100% specific, as you told, I strongly recommend to change this statement in the manuscript. At this moment you are able to inform that you have a Knockdown mouse and not a Knockout mouse. It does not impact the overall conclusion of your manuscript, since you have significant attenuation of your target protein, in a similar fashion as it is observed in the protein levels in the patients (S1C figure). In this scenario you have to rewrite several sentences in the manuscript.

Also, I would advise you to carefully check your supplemental materials, figures and tables. Several tables are out of format and are impossible to understand. Also, some figure legends do not properly describe your figure. Please, recheck them.

Figure S1C. Please, indicate in your WB representative image who is the patient, parent, and control.

Figure 4H and 4J. What do you mean by influence in the y axis?
I think it is fluorescence.

Your S3 table is not properly present.
Please, recheck.

Lines 173-177

Baseline beating frequency was measured.
Please, change your text.

Also you measured the velocity of contraction and relaxation. Change your text accordingly

Also you mention that contraction amplitude was not different in Cth and shTECRL but it was in Knock down hiPSC-CMs. However, in your figure legend there is no description about the data shown in figure 6C. Please better describe your result in your figure legend and text.

From your table S5 it is not possible to conclude anything.
Please, provide a better presentation of your data.

Correct line 211 and 212 to
Mice exhibited significant heart contraction defects in eight weeks, even though the mouse heart was normal..."

lines 218, correct to
overexpression of TECRL enhances maximal mitochondrial
respiratory capacity and this increase is....

lines 232 , change to
TECRL deficient cardiomyocytes have reduced expression of RYR2...

line 243, change to
TECRL expression is restricted to..

Lines 247 , change to
We showed that loss of Tecrl
results in altered Nrf2 as
as well as Fas expression.

Surface electrocardiogram.
Please, provide the ECG derivation
used for analysis.

For isolated mice cardiomyocytes experiments,
please inform if left, right, or both ventricles were used.
As it is written, it is not clear.
Also, for mitochondria experiments method (line 368) inform the
heart tissue used (left ventricle, right ventricle, both?)

For Fluorescent detection of ROS and Mitosox method
Please, describe the region of the heart used for analysis in the heart.
Right ventricle, left ventricle, both?

Reviewer #2 (Remarks to the Author):

The authors have greatly improved the manuscript and raised all my queries.
However, I found some issues found in the new text/figures that need to be solved:

1) ECG data: Authors refer to mouse surface electrocardiograms from another unpublished manuscript (CVR-2021-1489, Cuilan Hou, and Tingting Xiao) in Cardiovascular Research. If the authors decide to show representative traces, they must show the respective quantitative data detailing premature ventricular beats and ventricular tachycardia (VT) frequencies before and after the catecholaminergic stress. In addition, considering that lower heart rates predispose to arrhythmias (DAD-induced ectopic beats), please, provide average heart rate under anesthesia in the four groups. Importantly, ECG representative traces in Fig 2C-D show two different 50 ms scales. Please correct.

2) Lines 194-197:

- "...ATP synthase subunit beta, mitochondrial (ATPB)..." . Is this ok? Or is it subunit beta of mitochondrial ATP synthase?.

- Change "contaction" by "contact".

- Please, clarify the expression "was connected". Do you mean covalent bond? interaction? or what?

3) Consider rephrasing the following sentence (lines 204-209). It's not clear.

"Meanwhile, based on the cardiac RNA-seq, we observed that apoptosis-related proteins observed in the KEGG pathway analysis (Fig. 3C), we also found that apoptosis inducing factor (AIF) and cytochrome C (Cyc) were released from mitochondria into the cytoplasm after 60 nM siTECRL infection followed by immunofluorescent staining (Fig. 7F) and western blotting (Fig 7. G) in H9C2 cells."

4) Line 217: "...enen though" to "...even though"

5) What does "Cth" stands for? Control? If so, please use the conventional abbreviature "Ctl" instead of "Cth".

Review report and point to point replies:

Reviewer #1 (Remarks to the Author):

From Supplementary Table 1 it is really difficult to understand the result. Most importantly, it is not possible, as it is shown, to conclude that your KO mouse is indeed KO. Please, better describe your result. Also, at this stage I'm not 100% sure that your mice are indeed KO.

Since your antibody is not 100% specific, as you told, I strongly recommend to change this statement in the manuscript. At this moment you are able to inform that you have a Knockdown mouse and not a Knockout mouse. It does not impact the overall conclusion of your manuscript, since you have significant attenuation of your target protein, in a similar fashion as it is observed in the protein levels in the patients (S1C figure). In this scenario you have to rewrite several sentences in the manuscript.

Response: Thanks for your advice. We have rewritten in the reversion manuscript. We are sorry for the poor description of Supplementary Table 1 and we have rewritten in the reversion manuscript. We purchased a new TECRL antibody (SRD5A2L2 (TECRL) (C-term) rabbit polyclonal antibody, AP54042PU-N) and found that the expression of TECRL was significantly down-regulated by western blotting (Fig 1). See line 780-781, page 31.

Also, I would advise you to carefully check your supplemental materials, figures and tables. Several tables are out of format and are impossible to understand. Also, some figure legends do not properly describe your figure. Please, recheck them. Figure S1C. Please, indicate in your WB representative image who is the patient, parent, and control. Figure 4H and 4J. What do you mean by influence in the y axis? I think it is fluorescence. Your S3 table is not properly present. Please, recheck.

Response: Thanks for your advice. We are sorry for the poor description and have rewritten in the reversion manuscript.

We have added the WB representative image information in the revised Figure S1C. See line 801-802, page 38.

We have corrected in the revised Figure 4. See line 789-790, page 34.

We have uploaded in a suitable format and revised.

Lines 173-177 Baseline beating frequency was measured. Please, change your text. Also you measured the velocity of contraction and relaxation. Change your text accordingly.

Also you mention that contraction amplitude was not different in Cth and shTECRL but it was in Knock down hiPSC-CMs. However, in your figure legend there is no description about the data shown in figure 6C. Please better describe your result in your figure legend and text.

Response: Thanks for your advice. We are sorry for the poor description. We have revised in the reversion manuscript. Baseline beating frequency of hiPSC-CMs was measured. Statistical analysis showed no significant difference between the Cth and shTECRL group (Table S4). Musclemotion was used to determine the velocity of contraction and relaxation of the hiPSC-CMs beating in a two-second-long video. See line 175-179, page 8.

As for the contraction amplitude, there was no significance between the Ctl and shTECRL. We have rewritten in the reversion manuscript. See line 179-180, page 8. See line 795-796, page 36. The figure 6C figure legend was added in the reversion manuscript. See line 584-585, page 23.

From your table S5 it is not possible to conclude anything.

Please, provide a better presentation of your data.

Response: Thanks for your advice. We have uploaded in a suitable format and revised.

Correct line 211 and 212 to "Mice exhibited significant heart contraction defects in eight weeks, even though the mouse heart was normal..."

Response: Thanks for your advice. We have rewritten in the reversion manuscript. See line 211-212, page 10.

lines 218, correct to overexpression of TECRL enhances maximal mitochondrial

respiratory capacity and this increase is....

Response: Thanks for your advice. We have rewritten in the reversion manuscript.
See line 218-220, page 10.

lines 232, change to TECRL deficient cardiomyocytes have reduced expression of RYR2...

Response: Thanks for your advice. We have rewritten in the reversion manuscript.
See line 22, page 10.

line 243, change to TECRL expression is restricted to..

Response: Thanks for your advice. We have rewritten in the reversion manuscript.
See line 242-243, page 11.

Lines 247, change to We showed that loss of Tecrl results in altered Nrf2 as well as Fas expression.

Response: Thanks for your advice. We have rewritten in the reversion manuscript.
See line 246, page 11.

Surface electrocardiogram.

Please, provide the ECG derivation used for analysis.

Response: Thanks for your advice. We have rewritten in the reversion manuscript.
See line 323-332, page 14.

For isolated mice cardiomyocytes experiments, please inform if left, right, or both ventricles were used. As it is written, it is not clear.

Also, for mitochondria experiments method (line 368) inform the heart tissue used (left ventricle, right ventricle, both?)

Response: Thanks for your advice. We have added detailed information in the reversion manuscript. See line 350-351, page 15. See line 367-368, page 15.

For Fluorescent detection of ROS and Mitosox method. Please, describe the region of the heart used for analysis in the heart. Right ventricle, left ventricle, both?

Response: Thanks for your advice. We have added detailed information in the reversion manuscript. See line 423, page 17.

Reviewer #2 (Remarks to the Author):

1) ECG data: Authors refer to mouse surface electrocardiograms from another unpublished manuscript (CVR-2021-1489, Cuilan Hou, and Tingting Xiao) in Cardiovascular Research. If the authors decide to show representative traces, they must show the respective quantitative data detailing premature ventricular beats and ventricular tachycardia (VT) frequencies before and after the catecholaminergic stress. In addition, considering that lower heart rates predispose to arrhythmias (DAD-induced ectopic beats), please, provide average heart rate under anesthesia in the four groups. Importantly, ECG representative traces in Fig 2C-D show two different 50 ms scales. Please correct.

Response: Thanks for your advice. We have rewritten in the reversion manuscript. See line 101-103, page 5. See line 552-553, page 22. See line 783-784, page 32.

2) Lines 194-197: -“...ATP synthase subunit beta, mitochondrial (ATPB)...”. Is this ok? Or is it subunit beta of mitochondrial ATP synthase?.

-Change “contaction” by “contact”.

- Please, clarify the expression “was connected”. Do you mean covalent bond? interaction? or what

Response: Thanks for your advice. We are sorry for the poor description. We have rewritten in the reversion manuscript. ATP synthase subunit beta, mitochondrial (ATPB) is right.

We are sorry for the poor description. We have changed “was connected” to “was contacted”. See line 192-193, page 8.

Based on the mass spectrometry of TECRL co-immunoprecipitation (Table S6), we observed that ATP synthase subunit beta, mitochondrial (ATPB) was contacted with

TECRL, and the contact was significantly increased after TECRL overexpression (Fig. 7C).

3) Consider rephrasing the following sentence (lines 204-209). It's not clear.

“Meanwhile, based on the cardiac RNA-seq, we observed that apoptosis-related proteins observed in the KEGG pathway analysis (Fig. 3C), we also found that apoptosis inducing factor (AIF) and cytochrome C (Cyc) were released from mitochondria into the cytoplasm after 60 nM siTECRL infection followed by immunofluorescent staining (Fig. 7F) and western blotting (Fig 7. G) in H9C2 cells.”

Response: Thanks for your advice. We are sorry for the poor description. We have rewritten in the reversion manuscript. See line 200-203, page 8-9.

Apoptosis-related genes were altered via the above cardiac RNA-seq (Fig. 3C). We further noticed that apoptosis inducing factor (AIF) and cytochrome C (Cyc) were released from mitochondria into the cytoplasm (60 nM siTECRL) via immunofluorescent staining (Fig. 7F) and western blotting (Fig 7. G) in H9C2 cells.

4) Line 217: “...enen though” to “...even though”

Response: Thanks for your advice. We have rewritten in the reversion manuscript.

5) What does “Cth” stands for? Control? If so, please use the conventional abbreviature “Ctl” instead of “Cth”.

Response: Thanks for your advice. We have rewritten in the manuscript.

Reviewers' comments:

Reviewer #1 (Remarks to the Author):

Authors improved the manuscript. However, I still have two comments.

Please, keep the same bar color scheme along the manuscript. For some figures CTR bar is white, for others it is black. The same for the KO. For some figures it is black for others it is blue. Keep the same style along the manuscript.

Figure 6C. The representative trace show both, velocity of contraction and relaxation. However, in analysis you show only velocity of contraction. Please, correct the y axis accordingly

Reviewer #2 (Remarks to the Author):

The authors have improved the manuscript, but I'm still concerned about ECGs.

-ECG time scales are still duplicated (one as an inset in the ECG, and another below the ECG), please eliminate the extra scales.

-Line 338 (Surface electrocardiogram): I am wondering why the authors changed the ECG recording from a non-invasive method to a mild invasive method (needles). Please explain the reason for this change. If there is a change in the ECG method, why the ECG are the same?

-Line 338: Please, correct electrodes in "...from needle electrons..."

-Authors refer "multiple premature ventricular beats" in the Tecrl KO mice, but "few" ventricular events in WT mice (Fig. 2C-D). Please provide a quantification of premature ventricular beats (PVB), as requested in the previous revision. In addition, please show a PVB representative trace.

-Furthermore, lower respiratory rate and loss of toe-press reflex must imply a high degree of anesthesia, and that must include a lower heart rate. Please provide "average heart rate under anesthesia in the four groups" as requested in the previous revision.

Line 197: "was in contact" instead of "was contacted"

Please rephrase lines 207-208: "Apoptosis-related genes were altered via the above cardiac RNA-seq (Fig. 3C)." These genes were not altered via the RNA-seq.

Figure 6. The authors have changed average data in panel 6C following #1 reviewer's suggestion. What does it mean velocity of contraction and relaxation (Y-axis legend) as a unique value? The velocity of contraction must have a value, and relaxation another value. Or the results are only velocity of contraction? In addition, no units were provided.

Line 194: "in agreement" was right.

Review report and point to point replies:

Reviewers' comments:

Reviewer #1 (Remarks to the Author):

Authors improved the manuscript. However, I still have two comments.

Please, keep the same bar color scheme along the manuscript. For some figures CTR bar is white, for others it is black. The same for the KO. For some figures it is black for others it is blue. Keep the same style along the manuscript.

Response: Thanks for your advice. We have rewritten in the reversion manuscript. See line 790-803, page 34-38.

Figure 6C. The representative trace show both, velocity of contraction and relaxation. However, in analysis you show only velocity of contraction. Please, correct the y axis accordingly

Response: Thanks for your advice. We have rewritten in the reversion manuscript. See line 796-797, page 36.

Reviewer #2 (Remarks to the Author):

The authors have improved the manuscript, but I'm still concerned about ECGs.

-ECG time scales are still duplicated (one as an inset in the ECG, and another below the ECG), please eliminate the extra scales.

Response: Thanks for your advice. We have rewritten in the reversion manuscript. See line 784-785, page 32.

-Line 338 (Surface electrocardiogram): I am wondering why the authors changed the ECG recording from a non-invasive method to a mild invasive method (needles). Please explain the reason for this change. If there is a change in the ECG method, why the ECG are the same?

-Line 338: Please, correct electrodes in "...from needle electrons..."

Response: Agree. We are sorry for the mistake. We have rewritten in the reversion manuscript. See line 325-326, page 14.

-Authors refer “multiple premature ventricular beats” in the Tecrl KO mice, but “few” ventricular events in WT mice (Fig. 2C-D). Please provide a quantification of premature ventricular beats (PVB), as requested in the previous revision. In addition, please show a PVB representative trace.

-Furthermore, lower respiratory rate and loss of toe-press reflex must imply a high degree of anesthesia, and that must include a lower heart rate. Please provide “average heart rate under anesthesia in the four groups” as requested in the previous revision.

Response: Agree. Thanks for your suggestion. We have rewritten in the reversion manuscript. See line 101-105, page 5. See line 552-554, page 22. See line 784-785, page 32.

Line 197: “was in contact” instead of “was contacted”

Response: Agree. Thanks for your suggestion. We have rewritten in the reversion manuscript. See line 194, page 8.

Please rephrase lines 207-208: “Apoptosis-related genes were altered via the above cardiac RNA-seq (Fig. 3C).” These genes were not altered via the RNA-seq.

Response: Agree. Thanks for your suggestion. We have rewritten in the reversion manuscript. See line 202, page 8.

Figure 6. The authors have changed average data in panel 6C following #1 reviewer’s suggestion. What does it mean velocity of contraction and relaxation (Y-axis legend) as a unique value? The velocity of contraction must have a value, and relaxation another value. Or the results are only velocity of contraction? In addition, no units were provided.

Response: Thanks for your advice. We have rewritten in the reversion manuscript. See line 796-797, page 36.

Line 194: “in agreement” was right.

Response: Agree. Thanks for your suggestion. We have rewritten in the reversion

manuscript. See line 191, page 8.

REVIEWERS' COMMENTS:

Reviewer #2 (Remarks to the Author):

Minors comments.

-Regarding the comment about ECG monitoring system: Authors didn't give any explanation why they had changed the ECG recording method, as asked in the previous feedback ("I am wondering why the authors changed the ECG recording from a non-invasive method to a mild invasive method (needles). Please explain the reason for this change. If there is a change in the ECG method, why the ECG are the same?")

-Lines 796-797: No units provided in Y-axis of Figure 6C (it was asked in the previous feedback)

-Line 101-102: "The premature ventricular beats (PVB) was shown in Fig. 2E.". Please correct: ..."are shown"...

-Line 197: Please revise, still wrong.

I strongly suggest the authors proofread the manuscript by an English-native speaker.

REVIEWERS' COMMENTS:

Reviewer #2 (Remarks to the Author):

Minors comments.

-Regarding the comment about ECG monitoring system: Authors didn't give any explanation why they had changed the ECG recording method, as asked in the previous feedback ("I am wondering why the authors changed the ECG recording from a non-invasive method to a mild invasive method (needles). Please explain the reason for this change. If there is a change in the ECG method, why the ECG are the same?")

Response: We are sorry for the mistake. In the first version (COMMSBIO-21-1009A), we described the surface electrocardiogram in a non-invasive using the Mouse Monitor S from INDUS Technology. As the reviewer requested a more detailed description of the surface electrocardiogram method, we asked a student to fill in the details. Then, we mistakenly added a mild invasive method (needles) to the non-invasive method. For the surface electrocardiogram assay, we first used the powerlab from ADInstruments to detect the surface electrocardiogram in a mild invasive method (needles). Because the mild invasive method (needles) requires the experimenter to insert the electrode under the skin of the mouse, the insertion position and depth will affect the experimental data, so we adopt a non-invasive method to detect the surface electrocardiogram later. We apologize for confusing the two methods. The following Figure 1 showed the Mouse Monitor S from INDUS Technology, and the mouse on the test board. The surface electrocardiogram of mice can be detected by sticking the mice toes to the electrode, which is easy to operate and stable. In the manuscript (COMMSBIO-21-1009D), we used the Mouse Monitor S from INDUS Technology to detect mouse surface electrocardiogram. Lastly, we are sorry for the mistake. Thanks for your suggestion again.

Figure 1

-Lines 796-797: No units provided in Y-axis of Figure 6C (it was asked in the previous feedback)

Response: We are sorry for the mistake. Thanks for your advice. We have rewritten in the reversion manuscript. See line 794-795, page 36.

We analyzed the velocity of contraction according to the manufacturer’s protocol (Luca Sala, et al., *Circ Res.* 2018). Luca Sala, et al., showed the principle of using pixel intensity difference as a measure of displacement and as a measure of displacement velocity (Figure 2A-B).

Reference: MUSCLEMOTIONA Versatile Open Software Tool to Quantify Cardiomyocyte and Cardiac Muscle Contraction In Vitro and In Vivo. *Circ Res.* 2018 Feb 2; 122(3): e5–e16.

Figure 2

-Line 101-102: “The premature ventricular beats (PVB) was shown in Fig. 2E.”. Please

correct: ...“are shown”...

Response: Thanks for your advice. We have rewritten in the reversion manuscript. See line 102-103, page 5.

-Line 197: Please revise, still wrong.

Response: Thanks for your advice. We have rewritten in the reversion manuscript. See line 199-200, page 8.

I strongly suggest the authors proofread the manuscript by an English-native speaker.

Response: Thanks for your advice. The language of this manuscript has been improved through two steps. First, the authors read the manuscript again and corrected the writing. Then it was sent to Elsevier Language Editing Services for further language polishing with American English Style. Please see the revised manuscript and attached certificate.

Certificate of Elsevier Language Editing Services

The following article was edited by Elsevier Language Editing Services:
"TECRL deficiency results in aberrant mitochondrial function in cardiomyocytes"

Authored by:
Cuilan Hou